# A VARIATIONAL APPROACH FOR GENERATIVE SPEECH LANGUAGE MODELING

## ABSTRACT

The success of large language models in text processing has inspired their adaptation to speech modeling. However, because speech is continuous and complex, it is often discretized into tokens derived from self-supervised speech models. These speech tokens typically focus on the linguistic aspects of speech and neglect its paralinguistic content. As a result, autoregressive models trained on these tokens may generate speech with suboptimal naturalness. Previous methods attempted to address this limitation by adding pitch features to speech tokens prior to autoregressive modeling. However, pitch alone cannot fully represent the range of paralinguistic attributes, and selecting the right features requires careful hand-engineering. To tackle this issue, we propose a variational approach that automatically learns to encode these continuous speech attributes to enhance the speech tokens. Our proposed approach eliminates the need for manual paralinguistic feature selection and extraction. Moreover, we demonstrate that our proposed approach maintains or improves speech language modeling performance and enhances the naturalness of generated speech compared to baseline approaches.

## 1 INTRODUCTION

Large language models (LLMs) have achieved tremendous success in text processing OpenAI (2024), offering new ways to interact with machines. This progress has motivated efforts to extend their capabilities to speech to enable more natural spoken interactions with machines. However, modeling speech presents unique challenges due to its continuous and complex nature. As a result, previous works (Lakhotia et al., 2021; Borsos et al., 2023; Maiti et al., 2024) tokenized speech into simpler discrete units to enable the application of language modeling techniques originally developed for text. However, these *speech tokens* are typically derived by performing $k$-means clustering on features extracted from self-supervised pre-trained speech models, such as HuBERT (Hsu et al., 2021). These models primarily capture the linguistic aspects of speech, such as phonetic information, while often overlooking paralinguistic features, such as prosody (Weston et al., 2021). As a result, training an autoregressive model solely with such speech tokens restricts the model's ability to fully capture and represent the diverse information encoded in speech.

To address this limitation, Kharitonov et al. (2022) augmented the tokens with extracted fundamental frequency ($F_0$, or pitch) to enable prosody-aware modeling. However, augmenting speech tokens with manually defined paralinguistic attributes can be inherently suboptimal. First, pitch alone cannot capture the full range of paralinguistic information encoded in speech. For instance, energy-related (e.g., loudness, zero-crossing-rate) and spectral-related (e.g., mel-frequency cepstral coefficients) features are also important paralinguistic features (Schuller et al., 2009; 2013; Eyben et al., 2015). Additionally, training an accurate pitch tracker introduces additional complexity (Kim et al., 2018).

Instead of relying on hand-engineered paralinguistic features, we propose an approach to learning these features directly from the input signal, within an autoregressive framework. These learned features are optimized to simultaneously: 1) reconstruct the input speech and 2) enhance the autoregressive modeling process. Our approach allows the learned features to complement discrete speech tokens, removing the need for pre-extracted paralinguistic features as required in previous methods. As a result, our method generates more natural-sounding speech compared to baseline models, without sacrificing the meaningfulness of the syntheses.

## 2 PRELIMINARIES

In this work, we work on mel-spectrogram, and consider vocoding, the act of turning mel-spectrogram back to raw waveform, as a problem that has already been addressed. We denote the mel-spectrogram as $\mathbf{X} = (x_t \in \mathbb{R}^{d_x})_{t=1}^T$, where $d_x$ represents the number of filter-banks, $T$ is the total number of time frames in the spectrogram, and $x_t$ is the frame at time $t$. We use $\mathbf{X}_{i:j}$ to denote the sub-sequence $(x_t)_{t=i}^j$, and define $\mathbf{X}_{1:0} = \emptyset$. Our goal is to model $p(\mathbf{X})$ using a generative approach.

**Token-based Speech Language Model** We describe the general framework of speech language models that rely on the use of speech tokens, as seen in works like Lakhotia et al. (2021); Borsos et al. (2023); Maiti et al. (2024). This approach consists of three components: a speech tokenizer, an autoregressive model, and a decoder. The speech tokenizer maps $\mathbf{X}$[1] to a sequence of discrete speech tokens $\mathbf{Z}^d = (z_t^d \in \mathbb{N}_k)_{t=1}^T$, where $\mathbb{N}_k = \{1, 2, \dots, k\}$, and $k$ is the vocabulary size of the speech tokens. We use $p(\mathbf{Z}^d \mid \mathbf{X})$ to denote the implicit distribution of the pre-trained speech tokenizer. The autoregressive model, parameterized by $\psi$, models the probability of token sequences $\mathbf{Z}^d$ as $p_\psi(\mathbf{Z}^d) = \prod_{t=1}^T p_\psi(z_t^d \mid \mathbf{Z}_{1:t-1}^d)$. Finally, the decoder, parameterized by $\theta$, is trained to convert $\mathbf{Z}^d$ back to $\mathbf{X}$ by modeling $p_\theta(\mathbf{X} \mid \mathbf{Z}^d)$. However, this framework is limited to the speech tokens $\mathbf{Z}^d$, which primarily capture linguistic information and ignores paralinguistic information. As a result, the decoder $\theta$ may struggle with accurate reconstruction, and the autoregressive model $\psi$ can have difficulty incorporating paralinguistic information. To address this limitation, we propose to incorporate the variational autoencoder framework to learn continuous features to complement $\mathbf{Z}^d$.

**Variational Autoencoder (VAE)** Latent variable models introduce unobserved latent variables $\mathbf{Z}^c = (z_t^c \in \mathbb{R}^{d_z^c})_{t=1}^T$ that influence the observed variable $\mathbf{X}$. $d_z^c$ is the dimension of each $z_t^c$, and is a hyper-parameter chosen prior to training. In a VAE, the likelihood of the observed data given the latent variable, $p_\theta(\mathbf{X} \mid \mathbf{Z}^c)$, is modeled by a neural decoder, parameterized by $\theta$. The variational posterior, $q_\phi(\mathbf{Z}^c \mid \mathbf{X})$, is modeled by a neural encoder, parameterized by $\phi$. Using this modeling setup, the log-likelihood of the data, $\log p_\theta(\mathbf{X})$, can be written as:

$$\underbrace{\mathbb{E}_{\mathbf{Z} \sim q_\phi(\mathbf{Z}^c|\mathbf{X})}[\log p_\theta(\mathbf{X} \mid \mathbf{Z}^c)] - D_{KL}(q_\phi(\mathbf{Z}^c \mid \mathbf{X})||p(\mathbf{Z}^c))}_{\mathcal{O}_{ELBO}} + D_{KL}(q_\phi(\mathbf{Z}^c \mid \mathbf{X})||p_\theta(\mathbf{Z}^c \mid \mathbf{X})), \quad (1)$$

where $D_{KL}$ is the Kullback–Leibler (KL) divergence between two distributions, and $p(\mathbf{Z}^c)$ is a fixed prior distribution (usually a Gaussian). In Equation 1, $\mathcal{O}_{ELBO}$ is known as the evidence lower bound (ELBO), which provides a lower bound for $\log p_\theta(\mathbf{X})$ since $D_{KL}(q_\phi(\mathbf{Z}^c \mid \mathbf{X})||p_\theta(\mathbf{Z}^c \mid \mathbf{X}))$ is always non-negative. Therefore, instead of maximizing $\mathbb{E}_{\mathbf{X}}[\log p_\theta(\mathbf{X})]$ directly, the VAE maximizes the tractable lower bound $\mathbb{E}_{\mathbf{X}}[\mathcal{O}_{ELBO}]$. Here, we refer to the learned continuous latent $\mathbf{Z}^c$ from the VAE as the *variational features*.

## 3 PROPOSED FRAMEWORK

Figure 1 provides an overview of our proposed framework. This section is organized as follows: Section 3.1 introduces our setup that combines a VAE with an autoregressive model for the latent variables. Section 3.2 describes how we integrate speech tokens into the framework. Section 3.3 discusses how to balance the different loss terms that arise in our setup. Section 3.4 describes the use of normalizing flows to improve the expressive power of the autoregressive prior. Finally, Section 3.5 introduces the diffusion decoder and the utterance encoder used in the framework.

### 3.1 VARIATIONAL AUTOENCODER WITH AUTOREGRESSIVE PRIOR

Our method starts by modeling the prior of the VAE with a trainable autoregressive model $p_\psi(\mathbf{Z}^c) = \prod_{t=1}^T p_\psi(z_t^c \mid \mathbf{Z}_{1:t-1}^c)$. The concept of a VAE with a parameterized autoregressive prior has been explored in previous works (Vahdat & Kautz, 2020; Zhu et al., 2020) within the vision domain. Our work adopts this to speech continuation and further integrates discrete token-based models to enhance

---

[1]Speech tokenizers can operate on mel-spectrograms or directly on raw waveforms.

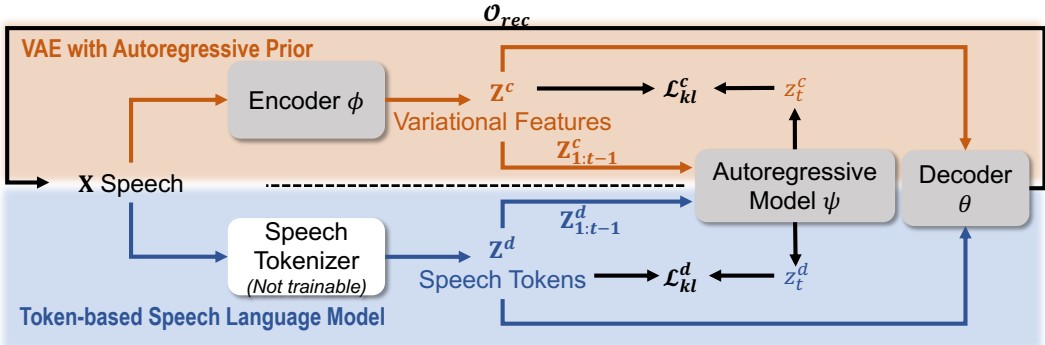

Figure 1: Overview of our proposed approach. Our method integrates the token-based speech language model (outlined in Section 2, represented by the lower shaded region) with a variational autoencoder (VAE with autoregressive prior, shown in the upper shaded region). This setup allows the model to learn variational features $\mathbf{Z}^c$ that complement the pre-extracted discrete speech token $\mathbf{Z}^d$. In our proposed joint setup, the variational features $\mathbf{Z}^c$ are trained to 1) reconstruct speech $\mathbf{X}$ alongside $\mathbf{Z}^d$ (by maximizing $\mathcal{O}_{rec}$); 2) facilitate the prediction of the next speech token $z_t^d$ (by minimizing $\mathcal{L}_{kl}^d$); 3) support the sequential prediction of the variational features themselves (by minimizing $\mathcal{L}_{kl}^c$).

synthesis naturalness. We use a diagonal Gaussian distribution to model the variational posterior, where the statistics are predicted by a neural network:

$$q_\phi(z_t^c \mid \mathbf{X}) = \mathcal{N}(z_t^c, \mu_\phi(\mathbf{X}, t), \sigma_\phi(\mathbf{X}, t)). \tag{2}$$

Since each $z_t^c$ is conditionally independent given $\mathbf{X}$, we can express the posterior as: $q_\phi(\mathbf{Z}^c \mid \mathbf{X}) = \prod_{t=1}^T q_\phi(z_t^c \mid \mathbf{X})$. With this decomposition, and the parameterized autoregressive prior, the $\mathcal{O}_{ELBO}$ in Equation 1 can be further derived[2] into:

$$\mathcal{O}_{ELBO} = \underbrace{\mathbb{E}_{\mathbf{Z}^c \sim q_\phi(\mathbf{Z}^c \mid \mathbf{X})}[\log p_\theta(\mathbf{X} \mid \mathbf{Z}^c)]}_{\mathcal{O}_{rec}} - \underbrace{\sum_{t=1}^T \mathbb{E}_{\mathbf{Z}_{1:t-1}^c} \left[ D_{KL}(q_\phi(z_t^c \mid \mathbf{X}) || p_\psi(z_t^c \mid \mathbf{Z}_{1:t-1}^c)) \right]}_{\mathcal{L}_{kl}^c}.$$

$$\tag{3}$$

By maximizing $\mathcal{O}_{ELBO}$, we maximize the first term, the reconstruction objective $\mathcal{O}_{rec}$, and minimize the second term, the variational feature prediction loss $\mathcal{L}_{kl}^c$. We note that training a model to maximize Equation 3 is feasible without incorporating discrete speech tokens $\mathbf{Z}^d$. This token-free approach is also depicted as the upper shaded region in Figure 1 (VAE with Autoregressive Prior), and its properties are explored in Section 5.

## 3.2 INCORPORATING THE SPEECH TOKENS

With the VAE with autoregressive prior in place, we now integrate speech tokens $\mathbf{Z}^d$ into the framework. By using these tokens, the model no longer needs to encode as much phonetic information in $\mathbf{Z}^c$, allowing $\mathbf{Z}^c$ to focus on other continuous speech attributes. To this end, we introduce a joint latent variable $\mathbf{Z} = (z_t \in \mathbb{R}^{d_z^c} \times \mathbb{N}_k)_{t=1}^T$, where $z_t$ is the concatenation of $z_t^c$ and $z_t^d$. Given that $\mathbf{Z}^d$ and $\mathbf{Z}^c$ are conditional independent given $\mathbf{X}$, we can express the new variational posterior as: $q_\phi(\mathbf{Z} \mid \mathbf{X}) = q_\phi(\mathbf{Z}^c \mid \mathbf{X})p(\mathbf{Z}^d \mid \mathbf{X})$. Then, we model $p_\psi(z_t \mid \mathbf{Z}_{1:t-1}) = p_\psi(z_t^d \mid \mathbf{Z}_{1:t-1})p_\psi(z_t^c \mid \mathbf{Z}_{1:t-1})$, assuming conditional independence of $z_t^d$ and $z_t^c$ given the past generations. We further discuss this

---

[2]See Appendix A.1 for detailed derivation.

modeling assumption in Appendix J. This allows us to re-write[3] $\mathcal{O}_{ELBO}$ from Equation 1 as:

$$\mathcal{O}_{ELBO} = \underbrace{\mathbb{E}_{\mathbf{Z}^d \sim p(\mathbf{Z}^d|\mathbf{X}), \mathbf{Z}^c \sim q_\phi(\mathbf{Z}^c|\mathbf{X})}[\log p_\theta(\mathbf{X} \mid \mathbf{Z}^d, \mathbf{Z}^c)]}_{\mathcal{O}_{rec}} - \tag{4}$$

$$\underbrace{\sum_{t=1}^{T} \mathbb{E}_{\mathbf{Z}_{1:t-1}}[D_{KL}(q_\phi(z_t^c \mid \mathbf{X})\|p_\psi(z_t^c \mid \mathbf{Z}_{1:t-1}))]}_{\mathcal{L}_{kl}^c} - \underbrace{\sum_{t=1}^{T} \mathbb{E}_{\mathbf{Z}_{1:t}}[-\log p_\psi(z_t^d \mid \mathbf{Z}_{1:t-1})]}_{\mathcal{L}_{kl}^d}.$$

From Equation 4, our training objective $\mathcal{O}_{ELBO}$ consists of three terms: $\mathcal{O}_{rec}$, $\mathcal{L}_{kl}^c$, and $\mathcal{L}_{kl}^d$. $\mathcal{O}_{rec}$ is the *reconstruction objective*. Maximizing $\mathcal{O}_{rec}$ trains the decoder $\theta$ to reconstruct $\mathbf{X}$ from both $\mathbf{Z}^c$ and $\mathbf{Z}^d$, while encouraging the encoder $\phi$ to generate $\mathbf{Z}^c$ with helpful information to reconstruct $\mathbf{X}$. $\mathcal{L}_{kl}^c$ is the *variational feature prediction loss*. Minimizing $\mathcal{L}_{kl}^c$ trains the autoregressive model $\psi$ to predict the next variational feature $z_t^c$ and encourages the encoder $\phi$ to generate $\mathbf{Z}^c$ that is easier for $\psi$ to model. $\mathcal{L}_{kl}^d$ is the *speech token prediction loss*, which trains the autoregressive model $\psi$ to predict the next speech token given the previous $\mathbf{Z}^d$ and $\mathbf{Z}^c$.

## 3.3 BALANCING THE LOSS TERMS

In Equation 4, the terms $\mathcal{O}_{rec}$, $\mathcal{L}_{kl}^c$, and $\mathcal{L}_{kl}^d$ can work against each other. For instance, the encoder $\phi$ optimizes both $\mathcal{O}_{rec}$ and $\mathcal{L}_{kl}^c$. Maximizing $\mathcal{O}_{rec}$ encourages the variational features $\mathbf{Z}^c$ to encode more information about $\mathbf{X}$, while minimizing $\mathcal{L}_{kl}^c$ regularize $\mathbf{Z}^c$ to be simpler for the autoregressive model $\psi$ to predict. Similarly, optimizing $\mathcal{L}_{kl}^c$ and $\mathcal{L}_{kl}^d$ with the autoregressive model $\psi$ a multi-task learning scenario, where $\psi$ learns to predict two different objectives given the same input. Moreover, these terms may operate on different scales due to how the losses are computed, necessitating a balancing mechanism. As a result, inspired by Higgins et al. (2017), we introduce two scalars, $\beta$ and $\gamma$, to balance the loss terms as follows:

$$\mathcal{O}_{ELBO} = \mathcal{O}_{rec} - \beta \cdot \mathcal{L}_{kl}^c - \gamma \cdot \mathcal{L}_{kl}^d. \tag{5}$$

Here, $\beta$ functions similarly to the parameter used in $\beta$-VAE (Higgins et al., 2017). A larger $\beta$ favors a simple $p(\mathbf{Z}^c)$, while a smaller $\beta$ encourages the variational features $\mathbf{Z}^c$ to encode more information about $\mathbf{X}$. Larger $\gamma$ encourages the autoregressive model $\psi$ to prioritize accurate predictions of $\mathbf{Z}^d$ over $\mathbf{Z}^c$. In practice, we employ a linear warm-up strategy for $\beta$, increasing it from zero to its final value during the early stages of training. This approach, inspired by prior works on text generation (Bowman et al., 2016; Fu et al., 2019), helps mitigate posterior collapse. Empirically, we find that this strategy allows for higher values of $\beta$ without causing $\mathcal{L}_{kl}^c$ to collapse to zero.

## 3.4 TIME-WISE NORMALIZING FLOW

We employ a lightweight normalizing flow Rezende & Mohamed (2015) that is shared across time to improve the expressive power of the autoregressive prior $p_\psi(z_t^c \mid \mathbf{Z}_{1:t-1})$. Specifically, an invertible flow network $f_\psi$ maps each $z_t$ to a point in the Gaussian distribution, and sampling can be realized by running the network in reverse. By using the change of variables, we can write:

$$p_\psi(z_t^c \mid \mathbf{Z}_{1:t-1}) = \mathcal{N}(f_\psi(z_t^c), \mu_\psi(\mathbf{Z}_{1:t-1}), \sigma_\psi(\mathbf{Z}_{1:t-1})) \left| \det \frac{\partial f_\psi(z_t^c)}{\partial z_t^c} \right|, \tag{6}$$

where $\mu_\psi, \sigma_\psi$ are modeled by autoregressive neural networks (i.e., transformer). We choose affine coupling layers Dinh et al. (2017) as the backbone of our normalizing flow due to their simple implementation and efficient computation. We note that similar approaches using normalizing flows to enhance prior distributions have also been observed in Kim et al. (2021; 2020) for text-to-speech.

## 3.5 OTHER COMPONENTS

We describe the modeling of the our decoder $p_\theta(\mathbf{X} \mid \mathbf{Z})$ and the utterance encoder designed to capture static information. While these components are not the main focus of our study, they help ensure a fair comparison between different methods. We use these components for all methods in our experiments and focus on how changing the inputs to the autoregressive model affects performance.

---

[3]See Appendix A.2 for detailed derivation.

**Diffusion Decoder**   We model the decoder $p_\theta(\mathbf{X} \mid \mathbf{Z})$ with Denoising Diffusion Probabilistic Model (DDPM) (Ho et al., 2020). We choose DDPM due to its flexibility in modeling complex distributions. We condition the diffusion process on $\mathbf{Z}$. For back-propagation through the encoder $\phi$, we use the reparameterization trick (Kingma & Welling, 2019) to sample from $q_\phi(\mathbf{Z}^c \mid \mathbf{X})$, and combine it with embedded speech tokens $\mathbf{Z}^d$. The outcome is then concatenated with each intermediate layer of the diffusion decoder for conditional diffusion. We train all diffusion decoders with 1000 DDPM steps.

**Utterance Encoder**   Static features, such as speaker information and recording environments, often vary little across a given utterance. In our current modeling approach, this static information would be redundantly encoded at each time step. To address this issue, we introduce an additional utterance-level feature encoder that encourages $\mathbf{Z}$ to focus on time-varying signals. Specifically, we randomly segment a portion of the mel-spectrogram $\mathbf{X}$ and feed it to the utterance encoder to produce an utterance-level embedding. This embedding is then concatenated with $\mathbf{Z}$ before being provided to the diffusion decoder. The utterance encoder is trained end-to-end with the entire system.

## 4 EXPERIMENTAL SETUP

### 4.1 DATASETS

We use two datasets in our experiments: LibriSpeech (Panayotov et al., 2015) and Libri-light (Kahn et al., 2020), consisting of audiobooks narrated in English. LibriSpeech contains 960 hours of speech, while Libri-light contains 60k hours of speech. For speech token extraction, we follow Hassid et al. (2023); Maiti et al. (2024) and use tokens derived from HuBERT representations (Hsu et al., 2021). We use the official HuBERT checkpoints, pre-trained on LibriSpeech[4] and Libri-light[5]. We run $k$-means clustering with $k = 200$ on the output of the last transformer layer of HuBERT using 10% of data randomly sampled from the training set. We pick $k = 200$ after testing values from $\{50, 200, 1000\}$ and choosing the one that produced the best language modeling performance The result is also consistent with Maiti et al. (2024). More details on the choice of $k$ are provided in Appendix F.

### 4.2 METHODS

We compare our proposed approach to methods that use only speech tokens in the autoregressive model, as well as methods that use speech tokens with added pitch features in the autoregressive model. To ensure a fair comparison, we fix the autoregressive model architecture to be the same for all methods, varying only the input and output layers. We also use the same configuration for the diffusion decoder and utterance encoder across all methods. For the neural vocoder (i.e., mapping the mel-spectrogram back to waveform), we train HiFi-GAN (Kong et al., 2020) on LibriSpeech and use it for all of the methods. We leave the detailed configuration of model architectures in Appendix B. Below, we provide further details on the three approaches.

**Token-LM**   We adopt the token-based speech language model (described in Section 2) as our baseline, representing approaches such as Lakhotia et al. (2021); Borsos et al. (2023); Maiti et al. (2024), which apply only discrete speech tokens to the autoregressive model.

**Token-LM + Pitch**   In this baseline approach, we augment the speech tokens of token-based speech language model (described in Section 2) with log pitch features before passing them into the autoregressive model. The pitch features are extracted using CREPE (Kim et al., 2018). Additionally, we introduce a pitch regression task alongside the standard next-token prediction task, optimizing it with L1 loss. This method incorporates hand-engineered paralinguistic features, similar to the approach used by Kharitonov et al. (2022).

**Variational speech modeling approach (Proposed)**   This is our proposed approach introduced in Section 3. In this approach, we learn to extract variational features that supplement the speech tokens while jointly training the autoregressive model. The learned variational features are used by both

---

[4] https://huggingface.co/facebook/hubert-base-ls960

[5] https://huggingface.co/facebook/hubert-large-ll60k

the autoregressive model and the decoder. This approach removes the need for a hand-engineered paralinguistic feature selection and extraction. Additionally, we set our latent dimension $d_z^c = 4$. While we observed performance improvements with larger $d_z^c$, we opted for a smaller value to ensure a fairer comparison, as it results in less variation in parameter size. Our additional experiments on the latent dimension $d_z^c$ is in Appendix E.

For inference, we use temperature-based sampling similar to Lakhotia et al. (2021). Specifically, we set the temperature to $0.85$ for both speech tokens $\mathbf{Z}^d$ and continuous variational features $\mathbf{Z}^c$. For variational features, the temperature is the scalar multiplied to the standard deviation of the normal distribution in Equation 6 before sampling, as done in Kim et al. (2020). For the diffusion decoder, we use denoising diffusion implicit models (DDIM) from Song et al. (2021) with $\eta = 0.5$ and $100$ diffusion steps. Training details are provided in Appendix C.

## 4.3 EVALUATION METRICS

We evaluate the comparison methods on both reconstruction and speech continuation. The reconstruction metrics, introduced in Section 4.3.1, involve only the encoder-decoder pair and indicate how much information is preserved in the extracted representations. The remaining metrics focus on speech continuation, which is our primary objective, where the performance of the autoregressive model is also assessed.

### 4.3.1 OBJECTIVE METRICS

**Reconstruction Metrics**  We use $F_0$-RMSE, mel-ceptral distortion (MCD), and character error rate (CER) to measure the quality of the reconstructed signal. $F_0$-RMSE measures the root mean squared difference between the pitch contour of the ground-truth signal and the reconstructed one. We use CREPE Kim et al. (2018) to extract pitch and only consider the voiced parts of the signal when computing the difference. MCD measures the Euclidean distance between the 23 mel-cepstral coefficients (MCEPs) extracted from the ground-truth and reconstructed signals. For calculating CER, we use a pre-trained Whisper Radford et al. (2022) automatic speech recognition model.[6] We use the `dev-clean` and `dev-other` subsets of LibriSpeech for evaluating reconstruction. To ensure deterministic results, instead of sampling each $z_t^c$ from $q_\phi(z_t^c \mid \mathbf{X})$, we directly use the Gaussian mean $\mu_\phi(\mathbf{X}, t)$ from Equation 2. In practice, we observed that the stochastic noise of $q_\phi(z_t^c \mid \mathbf{X})$ has little effect on the reconstructed syntheses.

**ZeroSpeech Metrics**  We adopt the commonly-used metrics (Borsos et al., 2023; Hassid et al., 2023; Maiti et al., 2024) from the ZeroSpeech challenge (Nguyen et al., 2020): sWUGGY and sBLIMP to measure language capability objectively. For these two metrics, speech utterances are given in positive-negative pairs, with each model scoring both utterances. The model's accuracy is the percentage of instances where the positive example receives a higher score than the negative one. sWuggy measures if the model scores a real word higher than a phonetically similar non-word (e.g., "brick" v.s. "blick"). sBLIMP measures if a model scores a grammatically correct sentence higher than a similar but incorrect one (e.g., "the dogs sleep" vs. "the dog sleep"). Both metrics use text-to-speech to generate the examples. In line with Borsos et al. (2023), we evaluate sWUGGY using only words existing in LibriSpeech (referred as the "in-vocab" version). We use the test split for evaluation. See Appendix G for detailed description on how we estimate the scores for the methods.

**Perplexity Metric**  We use perplexity to evaluate the language modeling capability of our methods. Specifically, perplexity measures the average negative log-likelihood of a test sequence generated by the language model. We use the `dev-clean` and `dev-other` subsets of LibriSpeech for computing perplexity.

### 4.3.2 SUBJECTIVE METRICS

We use subjective human evaluations to assess the naturalness and meaningfulness of the generated speech. We randomly sampled 100 utterances from the LibriSpeech `dev-clean` and `dev-other` subsets, cropping the first three seconds to use as prompts. Each audio sample was rated by seven

---

[6]https://huggingface.co/openai/whisper-medium

Table 1: Results of speech reconstruction evaluation ($F_0$-RMSE, MCD, CER) for the models discussed in Section 4.2. The evaluation metrics are detailed in Section 4.3. '# Param.' refers to the number of parameters used during inference. All models were trained on the Libri-light dataset.

| Method | # Param. | $F_0$-RMSE($\downarrow$) | MCD($\downarrow$) | CER($\downarrow$) |
|---|---|---|---|---|
| Ground-truth | *n/a* | *n/a* | *n/a* | 2.35 |
| *Token-LM* | 219M | 43.90 | 7.55 | 10.19 |
| *Token-LM + Pitch* | 219M | 25.46 | 6.90 | 6.59 |
| *Proposed* | 221M | **16.56** | **5.43** | **4.35** |

Table 2: Results of speech continuation evaluation for the models discussed in Section 4.2. The evaluation metrics are detailed in Section 4.3. M-MOS refers to the meaningfulness mean opinion score. N-MOS refers to the naturalness mean opinion score. Both M-MOS and N-MOS are evaluated on speech continuation, which are presented along with $95\%$ confidence intervals. All models were trained on the Libri-light dataset.

| Method | sWUGGY($\uparrow$) | sBLIMP($\uparrow$) | Perplexity($\downarrow$) | M-MOS($\uparrow$) | N-MOS($\uparrow$) |
|---|---|---|---|---|---|
| Ground-truth | *n/a* | *n/a* | *n/a* | $3.92 \pm 0.08$ | $3.94 \pm 0.09$ |
| *Token-LM* | **61.75** | **58.31** | 1.42 | $3.24 \pm 0.09$ | $3.01 \pm 0.10$ |
| *Token-LM + Pitch* | 60.75 | 56.92 | 1.40 | $3.17 \pm 0.09$ | $2.92 \pm 0.11$ |
| *Proposed* | 60.48 | 56.56 | **1.39** | $\mathbf{3.35} \pm 0.09$ | $\mathbf{3.37} \pm 0.10$ |

annotators. For naturalness, annotators rated how human-like the generated speech sounded on a five-point Likert scale, where one corresponds to "Very unnatural" and five to "Very natural." For meaningfulness, they rated the grammar and content of the speech on a five-point Likert scale, where one corresponds to "Very Poor" and five to "Excellent." Additional details on the subjective evaluations are provided in Appendix D.

# 5 EXPERIMENTAL RESULTS

## 5.1 MAIN RESULTS

Tables 1 and 2 present the results for the three methods described in Section 4.2. Table 1 reports both objective and subjective metrics for speech reconstruction, while Table 2 provides the corresponding results for speech language modeling. We discuss our observations below.

**Speech generated by our proposed approach is more natural—both objectively and subjectively—compared to the speech generated from the baselines.** The results in Table 1 show that our proposed approach improves the reconstruction of the original signal, as measured by three objective metrics: $F_0$-RMSE, MCD, and CER. These findings highlight three key points: 1) discrete speech tokens alone are insufficient to capture all the components necessary for faithful reconstruction, 2) incorporating only pitch information is not enough, and 3) the learned variational features $\mathbf{Z}^c$ in our approach effectively complement the discrete speech tokens $\mathbf{Z}^d$, leading to better reconstruction of speech signal. Furthermore, the subjective results of speech continuation, as measured by naturalness mean opinion score (N-MOS) in Table 2, show that *the syntheses produced by our proposed approach exhibit higher naturalness compared to baselines.* This finding further substantiates our hypothesis that the variational features $\mathbf{Z}^c$ learned by our approach contributes to improved synthesis. Additionally, Table 1 provides comparison of the number of parameters for each method. The result indicates that the overhead of the proposed method is relatively small (< 1% of the total parameters), while still achieving noticeably better performance.

**Speech generated using our proposed approach preserves subjective meaningfulness (as measured by M-MOS) comparable to the baselines, even though it shows a slightly higher likelihood of syntactic or grammatical errors in the objective sWUGGY and sBLIMP scores.** The results

Table 3: Results showing the impact of varying the $\beta$ parameter (as described in Section 3.3) and the effect of removing discrete tokens from our proposed approach on both language modeling and speech reconstruction performance. The $\gamma$ parameter (as described in Section 3.3) for the proposed methods is fixed to 0.5. All models were trained on the LibriSpeech dataset.

| Method | $\beta$ | sWUGGY($\uparrow$) | sBLIMP($\uparrow$) | $F_0$-RMSE($\downarrow$) | MCD($\downarrow$) | CER($\downarrow$) |
|---|---|---|---|---|---|---|
| *Token-LM* | *n/a* | 67.32 | **52.46** | 35.41 | 6.23 | 5.40 |
| *Token-LM + Pitch* | *n/a* | 66.49 | 51.65 | 21.08 | 6.05 | 5.08 |
| | 0.03 | 65.56 | 51.12 | **16.76** | **5.19** | **5.06** |
| *Proposed* | 0.04 | 65.96 | 51.40 | 16.88 | 5.53 | 5.43 |
| | 0.05 | 66.46 | 51.77 | 17.20 | 5.75 | 5.45 |
| | 0.03 | 67.79 | 51.76 | 16.86 | 5.24 | 10.83 |
| *Proposed* ($-$tokens) | 0.04 | 69.33 | 51.85 | 17.47 | 5.48 | 13.02 |
| | 0.05 | **71.11** | 51.86 | 18.64 | 5.84 | 16.51 |

in Table 2 show that our proposed approach produces comparable or better syntheses, as reflected by its higher meaningfulness mean opinion score (M-MOS), compared to the baselines. This claim is further supported by our method achieving the lowest Perplexity, suggesting that it can better predict the next speech token when the variational features $\mathbf{Z}^c$ are present. However, we also observe that both our approach and *Token-LM + Pitch* result in lower sWUGGY and sBLIMP scores compared to *Token-LM*. We speculate that encoding excessive information might introduce noise, potentially degrading performance on linguistic tasks like sWUGGY and sBLIMP. For example, factors such as speech loudness, which have low correlation with phonetic content, could add variability that negatively impacts likelihood estimation in these tasks. However, it is important to note that while sWUGGY and sBLIMP evaluate the likelihood of generating syntactically and grammatically correct sentences, they don't necessarily reflect the perceived meaningfulness for human listeners. The results indicate that our approach preserves subjective meaningfulness, even if it has a higher chance of syntactic or grammatical errors according to objective metrics.

## 5.2 ADDITIONAL ANALYSIS

We explore two additional experiments in this section. First, we study the effect of varying the balancing hyper-parameters, $\beta$ and $\gamma$ (described in Section 3.3). Second, we evaluate the utility of speech tokens in our proposed approach by training a model that uses only variational features $\mathbf{Z}^c$. This removal corresponds to training with Equation 3 instead of training with Equation 4. The results from these additional experiments are presented in Tables 3 and 4, and we discuss our observations below.

**Varying $\beta$** Table 3 shows that, for reconstruction metrics ($F_0$-RMSE, MCD, CER), lower values of $\beta$ result in smaller errors, indicating better reconstruction. However, for the sWUGGY and sBLIMP metrics, performance decreases as $\beta$ increases. This finding aligns with our discussion in Section 3.3, where we discussed how lower $\beta$ values encourage better reconstruction, but make it harder for the autoregressive model to effectively model $\mathbf{Z}^c$.

**Removing the discrete speech tokens** Table 3 shows the impact of removing the discrete speech tokens from our proposed approach. We find that excluding speech tokens leads to a slight improvement in the sWUGGY metric compared to including them. However, this exclusion significantly worsens the CER, indicating poorer phonetic reconstruction. These results suggest that without discrete speech tokens, our approach struggles to effectively encode abstract phonetic information in the variational features ($\mathbf{Z}^c$) but still performs well on sWUGGY, possibly by leveraging other cues. One possible explanation is that the synthesized non-existent words in sWUGGY, being out-of-domain for the text-to-speech system, may exhibit subtle prosodic irregularities that our model is able to detect. On the other hand, the best reconstruction results are obtained when speech tokens are included, as removing them leads to worse reconstruction metrics. Finally, we observe that varying the $\beta$ parameter produces similar performance trends regardless of whether speech tokens are used.

Table 4: Results showing the impact of varying the $\gamma$ parameter (as described in Section 3.3) in our proposed approach on both language modeling and speech reconstruction performance. The $\beta$ parameter (as described in Section 3.3) is fixed to $0.04$. M-MOS denotes the meaningfulness mean opinion score, and N-MOS denotes the naturalness mean opinion score, both presented with $95\%$ confidence intervals. All models were trained on the Libri-light dataset. Due to space constraints, sBLIMP and MCD results are presented in Appendix H. sBLIMP shows a similar trend to sWUGGY, while MCD mirrors the trend observed in CER.

| $\gamma$ | sWUGGY($\uparrow$) | Perplexity($\downarrow$) | $F_0$-RMSE($\downarrow$) | CER($\downarrow$) | M-MOS($\uparrow$) | N-MOS($\uparrow$) |
|---|---|---|---|---|---|---|
| 0.5 | **60.48** | 1.39 | **16.56** | 4.35 | **3.35** $\pm$ 0.09 | **3.37** $\pm$ 0.10 |
| 1.0 | 59.41 | 1.34 | 17.06 | 4.05 | 3.13 $\pm$ 0.09 | 3.27 $\pm$ 0.10 |
| 2.0 | 58.19 | **1.32** | 17.41 | **3.75** | 3.02 $\pm$ 0.09 | 3.09 $\pm$ 0.10 |

**Varying $\gamma$**   Table 4 shows that increasing $\gamma$ leads to worse pitch reconstruction, as measured by $F_0$-RMSE, but improves CER. This result indicates that $\gamma$ governs the type of information captured in the variational feature $\mathbf{Z}^c$. With a higher $\gamma$, the system prioritizes the prediction of speech tokens. Therefore, the variational feature $\mathbf{Z}^c$ is encouraged to encode more phonetic information, resulting in lower CER and MCD. Conversely, a lower $\gamma$ encourages $\mathbf{Z}^c$ to focus more on encoding pitch-related information, as indicated by the lower $F_0$-RMSE. Then, we analyze the subjective measures and observe that both M-MOS and N-MOS favor a lower $\gamma$. We attribute the decline in performance to the increased difficulty of autoregressive generation of $\mathbf{Z}^c$. By increasing the weight of $\mathcal{L}_{kl}^d$, the model sacrifices its focus on minimizing $\mathcal{L}_{kl}^c$, which in turn compromises its ability to model $\mathbf{Z}^c$.

**Benefit of trainable encoder**   We compare the reconstruction metrics of Table 1 and Table 3. Interestingly, our proposed method benefits from more data, showing better reconstruction quality when trained on Libri-light than on LibriSpeech, whereas the baseline methods do not. For *Token-LM*, the reconstruction relies entirely on the quality of extracted speech tokens. If these tokens lack sufficient phonetic information, the decoder cannot reconstruct accurate content, even with more data. In contrast, our approach allows the encoder to extract necessary information directly from the input speech, leveraging additional data to improve generalizability of both the encoder and decoder.

# 6 RELATED WORK

Emerging speech language models typically use discrete speech tokens for autoregressive modeling. These tokens are often obtained by $k$-means clustering of features extracted from self-supervised pre-trained models (Hsu et al., 2021; Chen et al., 2022). Lakhotia et al. (2021) used discrete speech tokens for generative spoken language modeling (GSLM). subsequently, Kharitonov et al. (2022) enhanced this approach by incorporating pitch information alongside speech tokens as joint inputs to the autoregressive model. Our proposed approach improves upon this line of research by using a variational autoencoder to automatically learn paralinguistic speech attributes in conjunction with the autoregressive model. Borsos et al. (2023) proposed a two-stage approach for the decoder that used acoustic tokens (Zeghidour et al., 2022; Défossez et al., 2022). This type of framework is also widely used in text-to-speech systems (Wang et al., 2023; Chen et al., 2024). In contrast, our approach focuses on the joint modeling of linguistic and paralinguistic features by enhancing the inputs to the autoregressive model rather than improving the decoder.

Recently, a line of research has emerged focusing on improving speech language models through the integration of text-based models. Hassid et al. (2023) initialized their speech language model using a pre-trained text-based large language model (LLM). Similarly, Rubenstein et al. (2023); Maiti et al. (2024) expanded the vocabulary of pre-trained text-based LLMs by integrating discrete speech tokens.Building on this, Yang et al. (2023); Du et al. (2024) further explored multi-task training involving text-conditioned generative speech tasks, combining text and audio within a single LLM. We note that our proposed approach takes a different direction but can still be integrated with these approaches. For example, one could initialize the transformer in our autoregressive model using parameters from a text-based LLM.

# 7 CONCLUSION

In this work, we proposed an approach that combines a variational autoencoder with existing token-based speech language models. We conducted experiments to evaluate its effectiveness in terms of language capability and synthesis naturalness. Empirical evaluations suggest that our proposed approach, in contrast with other recent techniques, is capable of producing synthesis with better subjective meaningfulness and naturalness. Additionally, we examined the effects of the weights of different loss terms, $\beta$ and $\gamma$, on performance. Our findings indicate that $\beta$ governs the amount of information encoded from the mel-spectrogram into the variational feature, whereas $\gamma$ controls the type of information encoded within the variational feature.

# 8 LIMITATIONS AND FUTURE WORK

Our results indicate that the performance of our proposed approach is sensitive to the choice of hyperparameters $\beta$ and $\gamma$. In future work, we plan to explore automated methods for tuning these parameters. Additionally, this study does not examine how our method can be leveraged for other downstream speech tasks. To address this, we plan to evaluate our pre-trained autoregressive transformer on downstream tasks using the SUPERB benchmark (Yang et al., 2024). Finally, our model has a relatively small number of parameters and requires less training data compared to many existing frameworks (Hassid et al., 2023; Rubenstein et al., 2023). We plan to scale our methods to assess whether the same conclusions hold with increased computational resources and larger datasets.

**Broader Impact** We proposed an approach that improves the naturalness of speech language models without compromising their language proficiency, which can be leveraged by existing paradigms in this literature. While a model that generates more natural speech can enhance the user experience in conversational agents, it can also be exploited for harmful purposes, such as creating fake videos or conducting spam phone calls.

**Reproducibility Statement** We provide detailed information about the model architecture in Appendix B, and the training details in Appendix C. Additionally, we include the audio samples used in human listening tests (M-MOS and N-MOS), along with a detailed setup of these evaluations in Appendix D. All datasets used are available for research purposes. We plan to open-source our code upon acceptance of the paper.

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

# A MATHEMATICAL DERIVATIONS

## A.1 EQUATION 3

For notation simplicity, we drop the superscript $c$ of $\mathbf{Z}^c$ into $\mathbf{Z}$ in this proof.

With the parameterized prior, the modeling distribution of $\mathbf{X}$ now also depends on $\psi$:

$$p_{\theta,\psi}(\mathbf{X}) = \int p_\theta(\mathbf{X} \mid \mathbf{Z}) p_\psi(\mathbf{Z}) d\mathbf{Z},$$

$$p_{\theta,\psi}(\mathbf{Z} \mid \mathbf{X}) = \frac{p_{\theta,\psi}(\mathbf{X}, \mathbf{Z})}{p_{\theta,\psi}(\mathbf{X})} = \frac{p_\theta(\mathbf{X} \mid \mathbf{Z}) p_\psi(\mathbf{Z})}{p_{\theta,\psi}(\mathbf{X})}.$$

Following a similar proof in Kingma & Welling (2019):

*Proof.*

$$
\begin{aligned}
\log p_{\theta,\psi}(\mathbf{X}) &= \mathbb{E}_{\mathbf{Z} \sim q_\phi(\mathbf{z}|\mathbf{X})}[\log p_{\theta,\psi}(\mathbf{X})] \\
&= \mathbb{E}_{\mathbf{Z} \sim q_\phi(\mathbf{z}|\mathbf{X})}\left[\log\left[\frac{p_{\theta,\psi}(\mathbf{X}, \mathbf{Z})}{p_{\theta,\psi}(\mathbf{Z} \mid \mathbf{X})}\right]\right] \\
&= \mathbb{E}_{\mathbf{Z} \sim q_\phi(\mathbf{z}|\mathbf{X})}\left[\log\left[\frac{p_{\theta,\psi}(\mathbf{X}, \mathbf{Z}) q_\phi(\mathbf{Z} \mid \mathbf{X})}{p_{\theta,\psi}(\mathbf{Z} \mid \mathbf{X}) q_\phi(\mathbf{Z} \mid \mathbf{X})}\right]\right] \\
&= \mathbb{E}_{\mathbf{Z} \sim q_\phi(\mathbf{z}|\mathbf{X})}\left[\log\left[\frac{p_{\theta,\psi}(\mathbf{X}, \mathbf{Z})}{q_\phi(\mathbf{Z} \mid \mathbf{X})}\right]\right] + \mathbb{E}_{\mathbf{Z} \sim q_\phi(\mathbf{z}|\mathbf{X})}\left[\log\left[\frac{q_\phi(\mathbf{Z} \mid \mathbf{X})}{p_{\theta,\psi}(\mathbf{Z} \mid \mathbf{X})}\right]\right] \\
&= \mathbb{E}_{\mathbf{Z} \sim q_\phi(\mathbf{z}|\mathbf{X})}\left[\log\left[\frac{p_{\theta,\psi}(\mathbf{X}, \mathbf{Z})}{q_\phi(\mathbf{Z} \mid \mathbf{X})}\right]\right] + D_{KL}(q_\phi(\mathbf{Z} \mid \mathbf{X}) || p_{\theta,\psi}(\mathbf{Z} \mid \mathbf{X})).
\end{aligned}
$$

Therefore,

$$\mathcal{O}_{ELBO} = \mathbb{E}_{\mathbf{Z} \sim q_\phi(\mathbf{Z}|\mathbf{X})} \left[ \log \left[ \frac{p_{\theta,\psi}(\mathbf{X}, \mathbf{Z})}{q_\phi(\mathbf{Z} \mid \mathbf{X})} \right] \right]$$

$$= \mathbb{E}_{\mathbf{Z} \sim q_\phi(\mathbf{Z}|\mathbf{X})} \left[ \log \left[ \frac{p_\theta(\mathbf{X} \mid \mathbf{Z}) p_\psi(\mathbf{Z})}{q_\phi(\mathbf{Z} \mid \mathbf{X})} \right] \right]$$

$$= \mathbb{E}_{\mathbf{Z} \sim q_\phi(\mathbf{Z}|\mathbf{X})} [\log p_\theta(\mathbf{X} \mid \mathbf{Z})] + \mathbb{E}_{\mathbf{Z} \sim q_\phi(\mathbf{Z}|\mathbf{X})} \left[ \log \left[ \frac{p_\psi(\mathbf{Z})}{q_\phi(\mathbf{Z} \mid \mathbf{X})} \right] \right]$$

$$= \mathbb{E}_{\mathbf{Z} \sim q_\phi(\mathbf{Z}|\mathbf{X})} [\log p_\theta(\mathbf{X} \mid \mathbf{Z})] - D_{KL}(q_\phi(\mathbf{Z} \mid \mathbf{X}) || p_\psi(\mathbf{Z})).$$

With $q_\phi(\mathbf{Z} \mid \mathbf{X}) = \prod_{t=1}^T q_\phi(z_t \mid \mathbf{X})$, and $p_\psi(\mathbf{Z}) = \prod_{t=1}^T p_\psi(z_t \mid \mathbf{Z}_{1:t-1})$:

$$D_{KL}(q_\phi(\mathbf{Z} \mid \mathbf{X}) || p_\psi(\mathbf{Z})) = \mathbb{E}_{\mathbf{Z} \sim q_\phi(\mathbf{Z}|\mathbf{X})} \left[ \log \left[ \frac{q_\phi(\mathbf{Z} \mid \mathbf{X})}{p_\psi(\mathbf{Z})} \right] \right]$$

$$= \mathbb{E}_{\mathbf{Z} \sim q_\phi(\mathbf{Z}|\mathbf{X})} \left[ \log \left[ \frac{\prod_{t=1}^T q_\phi(z_t \mid \mathbf{X})}{\prod_{t=1}^T p_\psi(z_t \mid \mathbf{Z}_{1:t-1})} \right] \right]$$

$$= \sum_{t=1}^T \mathbb{E}_{\mathbf{Z} \sim q_\phi(\mathbf{Z}|\mathbf{X})} \left[ \log \left[ \frac{q_\phi(z_t \mid \mathbf{X})}{p_\psi(z_t \mid \mathbf{Z}_{1:t-1})} \right] \right]$$

$$= \sum_{t=1}^T \mathbb{E}_{\mathbf{Z}_{1:t-1}} [D_{KL}(q_\phi(z_t \mid \mathbf{X}) || p_\psi(z_t \mid \mathbf{Z}_{1:t-1}))],$$

where $\mathbf{Z}_{1:t-1} \sim \prod_{t=1}^T q_\phi(z_t \mid \mathbf{X})$. □

### A.2 EQUATION 4

*Proof.* Since $\mathcal{O}_{rec}$ is straightforward to derive from Equation 1 (decompose $\mathbf{Z}$ into $\mathbf{Z}^c$ and $\mathbf{Z}^d$), here we show how $\mathcal{L}_{kl}^c$ and $\mathcal{L}_{kl}^d$ are derived from the $D_{KL}(q_\phi(\mathbf{Z} \mid \mathbf{X}) || p_\psi(\mathbf{Z}))$ in Equation 1.

With $q_\phi(\mathbf{Z} \mid \mathbf{X}) = q_\phi(\mathbf{Z}^c \mid \mathbf{X}) p(\mathbf{Z}^d \mid \mathbf{X})$ and $p_\psi(z_t \mid \mathbf{Z}_{1:t-1}) = p_\psi(z_t^d \mid \mathbf{Z}_{1:t-1}) p_\psi(z_t^c \mid \mathbf{Z}_{1:t-1})$:

$$D_{KL}(q_\phi(\mathbf{Z} \mid \mathbf{X}) || p_\psi(\mathbf{Z}))$$

$$= \mathbb{E}_{\mathbf{Z}} \left[ \log \left[ \frac{q_\phi(\mathbf{Z} \mid \mathbf{X})}{p_\psi(\mathbf{Z})} \right] \right]$$

$$= \mathbb{E}_{\mathbf{Z}} \left[ \log \left[ \frac{q_\phi(\mathbf{Z}^c \mid \mathbf{X}) p(\mathbf{Z}^d \mid \mathbf{X})}{\prod_{t=1}^T p_\psi(z_t \mid \mathbf{Z}_{1:t-1})} \right] \right]$$

$$= \mathbb{E}_{\mathbf{Z}} \left[ \log \left[ \frac{q_\phi(\mathbf{Z}^c \mid \mathbf{X}) p(\mathbf{Z}^d \mid \mathbf{X})}{\prod_{t=1}^T p_\psi(z_t^c \mid \mathbf{Z}_{1:t-1}) p_\psi(z_t^d \mid \mathbf{Z}_{1:t-1})} \right] \right]$$

$$= \mathbb{E}_{\mathbf{Z}} \left[ \log \left[ \frac{q_\phi(\mathbf{Z}^c \mid \mathbf{X})}{\prod_{t=1}^T p_\psi(z_t^c \mid \mathbf{Z}_{1:t-1})} \right] \right] + \mathbb{E}_{\mathbf{Z}} \left[ \log \left[ \frac{p(\mathbf{Z}^d \mid \mathbf{X})}{\prod_{t=1}^T p_\psi(z_t^d \mid \mathbf{Z}_{1:t-1})} \right] \right]$$

$$= \sum_{t=1}^T \mathbb{E}_{\mathbf{Z}_{1:t-1}} [D_{KL}(q_\phi(z_t^c \mid \mathbf{X}) || p_\psi(z_t^c \mid \mathbf{Z}_{1:t-1}))] - \sum_{t=1}^T \mathbb{E}_{\mathbf{Z}_{1:t}} [\log p_\psi(z_t^d \mid \mathbf{Z}_{1:t-1})]$$

$$+ \mathbb{E}_{\mathbf{Z}}[\log p(\mathbf{Z}^d \mid \mathbf{X})]$$

Since $\mathbb{E}_{\mathbf{Z}}[\log p(\mathbf{Z}^d \mid \mathbf{X})]$ does not depends on any parameters, it can be dropped during optimization. □

## B  MODEL ARCHITECTURES

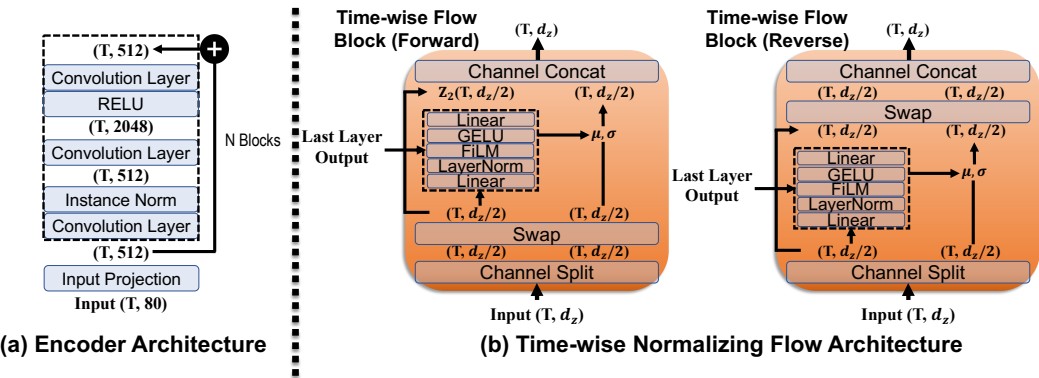

Figure 2: (a) Residual block architecture or the encoder $\phi$. (b) Model architecture for the time-wise normalization flow introduced in Section 3.4.

Table 5: Model configuration of the autoregressive transformer for training on LibriSpeech and Libri-light respectively. This configuration is shared for all comparing methods. 'feed-forward size' refers to the width of the feed-forward linear layer.

| Dataset | # of layers | # of heads | hidden size | feed-forward size |
|---|---|---|---|---|
| LibriSpeech | 4 | 8 | 512 | 2048 |
| Libri-light | 16 | 16 | 1024 | 4096 |

**Encoder** $q_\phi(\mathbf{Z} \mid \mathbf{X})$    We use a different number of residual blocks for the encoder. We use a kernel size of 7; the hidden dimensions used for all models are in Figure 2 (a). The architecture of the residual block is illustrated in Figure 2 (a). Finally, after 3 residual blocks, we apply another instance normalization, followed by separate linear heads to output the mean and log variance of Equation 2. We use the same size encoder for experiments on LibriSpeech and Libri-light. Instance Norm refers to instance normalization (Ulyanov et al., 2017).

**Autoregressive Transformer**    We follow the typical implementation of transformers with Post-LN (Xiong et al., 2020). We use RMSNorm (Zhang & Sennrich, 2019) and GELU activation (Hendrycks & Gimpel, 2023). We use ALiBi (Press et al., 2022) for relative positional encoding. We use different model sizes for LibriSpeech and Libri-light experiments, with the configuration summarized in Table 5. The same configuration is shared for all comparing methods.

**Time-wise Normalizing Flow**    The architecture of our time-wise normalizing flow is illustrated in Figure 2 (b). Here, $\mu$ and $\sigma$ are the mean and standard deviation that will be multiplied and added to the input. This part mainly follows the implementation from Dinh et al. (2017). The "Last Layer Output" in Figure 2 (b) refers to the output of the last transformer layer. "FiLM" refers to FiLM conditioning (Perez et al., 2018). "Swap" refers to swapping the two inputs in their channel order. We use 4 flow blocks for all experiments.

**Diffusion Decoder**    For our diffusion decoder $\theta$, we apply the same residual block as Figure 2 (a). However, here we have additional skip connections between output of residual blocks following the commonly-used U-Net architecture (Ronneberger et al., 2015). We encode the current diffusion step with Sinusoidal positional encoding, linear project it and add it to each time frame of the output of the first convolution layer in each of the residual blocks. For both datasets, we use 6 residual blocks, with the same hidden dimensions and kernel size as that of the encoder $\phi$.

**Utterance Encoder**    The utterance encoder consists of 3 blocks, where each block sequentially includes a convolution with stride 2 and kernel size 4, followed by instance normalization (Ulyanov et al., 2017) and RELU activation. The hidden size of the convolution layer is: 128, 256, 512. Afterward, a simple time-averaging is applied to the output to generate an utterance-level embedding.

Please listen to the computer-generated speech sample below and rate how well its grammar and content convey meaningful information. Focus on evaluating the grammar and the content, not the naturalness or quality of the speech.

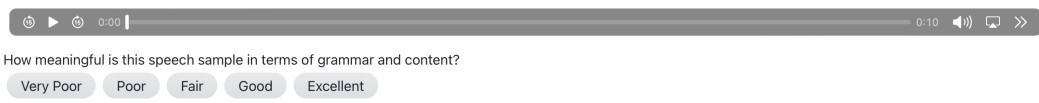

How meaningful is this speech sample in terms of grammar and content?

Very Poor   Poor   Fair   Good   Excellent

Figure 3: A screenshot of the Meaningfulness (M-MOS) assessment task, as the crowd-sourced rater sees it.

Please listen to the computer-generated speech sample below and rate how well its grammar and content convey meaningful information. Focus on evaluating the grammar and the content, not the naturalness or quality of the speech.

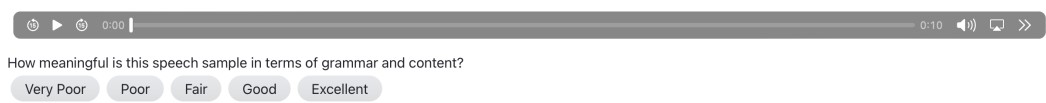

How meaningful is this speech sample in terms of grammar and content?

Very Poor   Poor   Fair   Good   Excellent

Figure 4: A screenshot of the Naturalness (N-MOS) assessment task, as the crowd-sourced rater sees it.

## C  TRAINING DETAILS

For training the model, we use the AdamW optimizer (Loshchilov & Hutter, 2019) with $\beta_1 = 0.9, \beta_2 = 0.98$. We use weight decay of 0.01 for LibriSpeech models and 0.1 for the Libri-light models. We trained the models with mixed precision. For Libri-light models, we use 2 L40S GPUs with gradient accumulation of step size 2. This makes the effective batch size 192. We trained for 600k update steps. We warm up $\beta$ from 0 to the final value in the first 30k update steps. It takes about 14 days to train the Libri-light models.

For LibriSpeech models, we discovered that methods involving discrete tokens suffers from early overfitting (but not in Libri-light). Therefore, we train these models (including our proposed approach) to only 100k steps. For the diffusion decoder of *Token-LM* and *Token-LM + pitch*, we separately train them to 500k steps, where we observe marginal improvement of loss functions between epochs. For pure variational approaches, we train to 400k steps as we did not observe overfitting. We use the same effective batch size on the 2 L40S GPUs but without gradient accumulation. For LibriSpeech models, we warm up $\beta$ from 0 to the final value in the first 20k update steps. It takes about 2 days to train for the 400k step models and less than 1 day for the 100k step models. For both models, we use an initial learning rate of $5e - 4$ and apply cosine learning rate decay to $5e - 5$.

For the input to the utterance encoder, we random crop the segment to be between 2 to 4 seconds. For diffusion model, we use L1 loss to predict the diffusion noise, and apply the cosine schedule for the diffusion noise variance.

## D  SUBJECTIVE EVALUATION

We use crowd-sourcing for subjective human evaluation on speech meaningfulness and naturalness. The recruited raters speaks English and were paid at least the minimum wage. We sample 100 prompts from LibriSpeech development subsets, crop the first 3 seconds, and feed to each model to produce a 10 seconds continuation (totally 13 seconds). The 100 prompts are the same for all comparing methods. Since we do not train our model to predict the end of speech, we observed that some synthesis ends earlier than 13 seconds. We use pre-trained voice activity detection from pyannote[7] to post-process the samples, removing trailing silences and non-speech that might affect evaluation.

In Figures 3 and 4, we provide screenshots of the what the raters see during evaluation. Raters are presented with a spoken utterance and are instructed them to rate its naturalness or meaningfulness

---

[7]https://huggingface.co/pyannote/voice-activity-detection

Table 6: Performance varying latent dimension $d_z^c$ on our proposed approach (without speech tokens). Models are trained on LibriSpeech.

| $d_z^c$ | sWUGGY($\uparrow$) | sBLIMP($\uparrow$) | F0-RMSE($\downarrow$) | MCD($\downarrow$) | CER($\downarrow$) |
|---|---|---|---|---|---|
| 4 | 69.33 | **51.85** | 17.47 | 5.48 | 13.02 |
| 16 | **73.49** | 51.69 | **16.68** | 5.37 | 7.80 |
| 64 | 73.25 | 50.91 | 17.37 | **5.35** | **7.79** |

Table 7: Comparison of model trained on different number of discrete tokens $k$. Models are trained on LibriSpeech.

| $k$ | sWUGGY($\uparrow$) | sBLIMP($\uparrow$) | $F_0$-RMSE($\downarrow$) | MCD($\downarrow$) | CER($\downarrow$) |
|---|---|---|---|---|---|
| 50 | 59.63 | **52.49** | 41.11 | 6.49 | 11.87 |
| 200 | **67.32** | 52.46 | 35.41 | 6.23 | 5.40 |
| 1000 | 65.11 | 50.99 | **32.60** | **5.99** | **4.48** |

on a five-point Likert scale, where 1 corresponds to very unnatural or meaningless and 5 corresponds to very natural or meaningful.

# E    DIMENSION OF THE LATENT VARIABLE $d_z^c$

Table 6 presents our results of increasing the latent dimension $d_z^c$. We perform the sweep on the variational approach without speech tokens for simplicity. From Table 6, we observe that increasing the latent dimension from 4 to 16 results in uniform improvements across the measures. However, further increasing the dimension from 16 to 64 leads to marginal degradation. We speculate that this performance plateau may arise from the difficulty normalizing flows face when modeling higher-dimensional distributions (Reyes-González & Torre, 2023).

# F    DISCRETE TOKEN VOCABULARY SIZE

Table 7 shows our evaluation results on speech token models (*Token-LM*) trained with varying $k$. Here, $k$ refers to number of clusters for the $k$-means clustering on obtaining the discrete token, which is equal to the vocabulary size of the discrete tokens. Our result is consistent with Maiti et al. (2024), which shows that $k = 200$ obtains the best sWUGGY score. The reconstruction metrics indicate that $k = 200$ provides a significant improvement over $k = 50$, whereas the increasing from $k = 200$ to $k = 1000$ yields only marginal gain. Interestingly, having larger $k$ seems to negatively impact sBLIMP. We speculate that the small vocabulary size ($k = 50$) is adequate for distinguishing word-level changes in sentences but insufficient for detecting subtle phonetic variations within words.

# G    SCORING sWUGGY AND sBLIMP

**Token-LM**    To obtain the scores for sWUGGY and sBLIMP for discrete speech token only models, we follow Borsos et al. (2023) and use the log-likelihood returned by the model normalized by the sequence length.

**Token-LM + pitch, proposed methods**    For methods that have additional inputs other than the discrete tokens, we only use the model's log-likelihood of the discrete tokens. We do not use the log-likelihood of the $\mathbf{Z}^c$, as we assume that the discrete tokens $\mathbf{Z}^d$ should contain all the information needed for sWUGGY and sBLIMP. In practice, we indeed observe that including the log-likelihood of the $\mathbf{Z}^c$ slightly lowers the score for our proposed method.

**Proposed w.o. token**    Since there are no discrete tokens involved in *Proposed w.o. token*, we directly use the log likelihood of $\mathbf{Z}^c$. The likelihood can be estimated by using Equation 6.

Table 8: Results showing the impact of varying the $\gamma$ parameter (as described in Section 3.3) in our proposed approach on sBLIMP and MCD. These measures are dropped in Table 4 due to space constraints.

| $\gamma$ | sBLIMP($\uparrow$) | MCD($\downarrow$) |
|---|---|---|
| 0.5 | **59.88** | 5.43 |
| 1.0 | 59.12 | 5.36 |
| 2.0 | 58.19 | **5.21** |

Table 9: Performance of speech emotion recognition models trained on different features. The features are extracted from models pre-trained on Libri-light using our proposed method.

| Method | Emotion Recognition (ACC, %) |
|---|---|
| *Tokens* | $57.46 \pm 1.59$ |
| *Variational Features* | $91.57 \pm 0.35$ |
| *Tokens + Variational Features* | $\mathbf{92.74} \pm 0.37$ |

Table 10: Performance of speaker identification models trained on different features. The features are extracted from models pre-trained on Libri-light using our proposed method.

| Method | Speaker Identification (ACC, %) |
|---|---|
| *Tokens* | $7.08 \pm 0.40$ |
| *Variational Features* | $63.41 \pm 0.43$ |
| *Tokens + Variational Features* | $63.13 \pm 0.45$ |
| *Utterance Embedding* | $\mathbf{94.06} \pm 0.32$ |

For *Proposed* and *Proposed w.o. token*, to ensure deterministic outcome, we again use the $\mu_\phi(\mathbf{X}, t)$ from Equation 2 directly as $\mathbf{Z}^c$, instead of sampling $\mathbf{Z}^c$ from $q_\phi(z_t \mid \mathbf{X})$.

## H SBLIMP AND MCD RESULTS FOR TABLE 4

Table 8 shows the remaining measures (sBLIMP and MCD) for Table 4. We observe that MCD follows the same trend as CER, while sBLIMP aligns with the trend observed in sWUGGY.

## I SIDE EXPERIMENTS ON INSPECTING LEARNED FEATURES

**Speech Emotion Recognition** We evaluate speech emotion recognition on the EmoV-DB Adigwe et al. (2018) dataset. We follow a 9:1 split on training and testing for the dataset. The dataset contains five emotion categories: amused, angry, neutral, disgust, and sleepiness. We train a classifier with the same structure to predict emotion categories based on different features. The experiments are repeated 20 times to report the mean and 95% confidence interval. From Table 9, we can observe that the variational features alone obtain significantly better performance compared to tokens, showcasing its capability of capturing paralinguistic information. Combining both tokens and variational features gives a slight improvement over using variational features alone.

**Speaker Identification** For speaker identification, we evaluate the performance on the VCTK Yamagishi et al. (2019) dataset, which consists of read English sentences, with 400 sentences each from 110 speakers. We again follow a 9:1 train-test split and repeat each run 20 times to report the mean and 95% confidence interval. We additionally evaluate our utterance embedding, which is designed to capture static utterance-level information (see Section 3.5). From Table 10, we can see that using tokens only results in poor speaker identification accuracy. With variational features, the classifier obtains improved accuracy. We attribute this improvement to the fact that speaking styles can be captured in the variational features to classify speakers. On the other hand, the utterance embedding

outperforms the other features in this task. These results support our claim that the utterance encoder encodes global speaker information while variational features capture local paralinguistic attributes.

## J    CONDITIONAL INDEPENDENCE ASSUMPTION OF $z_t^d$ AND $z_t^c$

In general, $\mathbf{Z}^c$ and $\mathbf{Z}^d$ are not independent, since the language content can imply the paralinguistic information, and vice versa. However, our modeling assumes only conditional independence. Specifically, the past generations $\mathbf{Z}_{1:t-1}$ are first passed through the autoregressive transformer $\psi$ to produce the intermediate representation $o_t = Transformer_\phi(\mathbf{Z}_{1:t-1})$. Then, two separate heads predict $z_t^c$ and $z_t^d$ based on $o_t$. This framework assumes that the transformer can learn $o_t$ such that $z_t^c$ and $z_t^d$ become conditionally independent given $o_t$. Given the transformer's modeling capacity, we believe it can extract shared information ($o_t$) between $z_t^c$ and $z_t^d$ from $\mathbf{Z}_{1:t-1}$, while delegating the distinct information to their respective heads.

