# OpenReview forum: "A Variational Approach for Generative Speech Language Modeling"
_ICLR.cc/2025/Conference — Submitted to ICLR 2025_

### Official Review · Reviewer_kqQ1 · 2024-10-18

**Soundness:** 3
**Presentation:** 3
**Contribution:** 2
**Rating:** 6
**Confidence:** 4

**Summary:**

This paper proposes a variational approach to speech-language modelling in contrast to traditional auto-regressive models. The aim is to capture information other than semantics.

**Strengths:**

1. Using the variational method for speech-language modelling is novel and useful. The exploration is very interesting and I like the idea.
2. The paper is well-written with adequate derivations for key loss functions.

**Weaknesses:**

1. My main concern is how this method is useful for more practical downstream tasks such as ASR, emotion or speaker recognition, to reflect that capturing the additional (mainly paralinguistic) information is useful. I strongly encourage the authors to conduct at least 2 of the above practical tasks using the variational speech LM and compare it to token-based speech LM to see if there are any potential benefits of using variational methods.
2. Experimental is conducted using LibriSpeech and Libri-light, which are datasets with quite small variabilities other than semantic information. I believe the variability remains in the speaker representation space, which is not explicitly reflected in the experimental design.

Together with the question below, I am not convinced that this paper has acceptance quality at the moment. However, given the idea is interesting, I would like to see how the authors would improve the paper during the rebuttal period, and promise to raise my score if my concerns are addressed.

**Questions:**

1. Why do we need to separate z^c and z^d? Are they really independent? I believe knowing z^c will tell you a lot about z^d, right? The authors are encouraged to explain this design choice further.

---

> ### Author Response · Authors · 2024-11-20
>
> > My main concern is how this method is useful for more practical downstream tasks such as ASR, emotion or speaker recognition, to reflect that capturing the additional (mainly paralinguistic) information is useful. I strongly encourage the authors to conduct at least 2 of the above practical tasks using the variational speech LM and compare it to token-based speech LM to see if there are any potential benefits of using variational methods.
>
> We thank the reviewer for their valuable feedback. In our **general response**, we include two experiments—emotion recognition and speaker identification—to demonstrate the information captured by our learned features. The experiments requested by the reviewer strengthen our claims by showing that the features our model learns encode paralinguistic information.
>
> > Experimental is conducted using LibriSpeech and Libri-light, which are datasets with quite small variabilities other than semantic information. I believe the variability remains in the speaker representation space, which is not explicitly reflected in the experimental design.
>
> We thank the reviewer for their comment regarding the evaluation dataset. We agree that the LibriSpeech and Libri-light datasets are not the most expressive in terms of emotion. However, we chose these datasets because they are **widely used in the speech continuation literature** [1,2,3,4], and prior works [1] have indicated that achieving natural synthesis for speech continuation on these datasets is **already a challenging task**. Our results demonstrate that our approach enhances the naturalness of the generated continuations for these datasets. Given this, we believe that if our method can improve synthesis on these datasets, it is likely to show even larger improvements on more expressive datasets.
>
> In addition, our new results in the general response above show that training on LibriSpeech and Libri-light yields models whose features are useful for two downstream tasks (emotion recognition and speaker identification) on **out-of-domain datasets**.
>
> [1] E. Kharitonov, et al., Text-free prosody-aware generative spoken language modeling, in Proceedings of the 60th Annual Meeting of the Association for Computational Linguistics (Volume 1: Long Papers), pp. 8666–8681, Dublin, Ireland, May 2022.
>
> [2] Z. Borsos, et al., AudioLM: A language modeling approach to audio generation, IEEE/ACM Transactions on Audio, Speech, and Language Processing, 31:2523–2533, 2023.
>
> [3] M. Hassid, et al., Textually Pretrained Speech Language Models, ArXiv, abs/2305.13009, 2024.
>
> [4] K. Lakhotia, et al., On Generative Spoken Language Modeling from Raw Audio, Transactions of the Association for Computational Linguistics, 2021.

---

> ### Author Response · Authors · 2024-11-20
>
> > Why do we need to separate $z^c$ and $z^d$? Are they really independent? I believe knowing $z^c$ will tell you a lot about $z^d$, right? The authors are encouraged to explain this design choice further.
>
> We separate $\mathbf{Z}^c$ and $\mathbf{Z}^d$ to model different aspects of speech. $\mathbf{Z}^d$ models the language content, which is inherently discrete, and we are able to leverage prior works (e.g., Hubert) to use speech tokens that are shown to capture it. On the other hand, we believe that the remaining information in speech is better modeled continuously, and we use a variational approach to learn the variational feature $\mathbf{Z}^c$.
>
> We agree with the reviewer that $\mathbf{Z}^c$ and $\mathbf{Z}^d$ should not be independent; knowing the language content implies its paralinguistic information, and vice versa. However, we would like to clarify that the modeling assumption in our paper is **less strict**; we assume only **conditional independence**. The past generations are first passed to the same autoregressive transformer to produce the representation, e.g., we can denote this as $o_t=Transformer_{\phi}(\mathbf{Z}_{1:t-1})$.
>
> Then, we have different heads predicting the next $z^c_t$ and $z^d_t$ conditioned on $o_t$.
>
> Therefore, the assumption of this framework is that the transformer is able to learn $o_t$ from the past generations $\mathbf{Z}_{1:t-1}$ so that the $z^c_t$ and $z^d_t$ are conditionally independent given $o_t$.
>
> Given the modeling power, we believe that it is possible that the transformer can learn to extract shared information ($o_t$) between $z^c_t$ and $z^d_t$ from $\mathbf{Z}_{1:t-1}$ and leave the distinct information to the distinct heads.
>
>
> We acknowledge that the most rigorous modeling choice would be to first predict $z^d_t$ from the discrete head of the autoregressive transformer and then condition the generation of $z^c_t$ based on the generated $z^d_t$. In this way, we don't need any assumption of independence. While this can be incorporated into the framework, we opted for a less rigorous modeling which requires less engineering effort on both training and inference.
>
> We also would like to note that **even with this less rigorous modeling, our proposed approach already outperforms the baselines**.
>
> We will modify Section 3.2 accordingly to include this discussion.

---

> > ### Comment · Reviewer_kqQ1 · 2024-11-20
> > **Thanks**
> >
> > Thank the authors for their responses and it resolved most of my concerns. I raised my score accordingly. However, I do not fully understand why these features are not suitable for ASR or AST which I am slightly concerned that might limit its wider applicability.

---

> > > ### Author Response · Authors · 2024-11-20
> > >
> > > We sincerely thank the reviewers for the response which inspired us to think about wider applications of our approach.
> > > As our primary focus is on utilizing the proposed modeling to enhance the naturalness of the speech continuation task, we have not thoroughly explored its potential for other downstream tasks.
> > >
> > > To evaluate our approach on downstream tasks, a promising method could be fine-tuning the autoregressive transformer on these tasks, treating our method as a pre-training objective. We are indeed curious about how this approach compare to discrete-based SSL speech foundation models, such as HuBERT [1], when evaluated on various speech tasks.
> > >
> > > That said, as the main focus of this paper remains generative speech modeling, we leave these investigations for future work.
> > > We will modify the limitation section (Section 8).
> > >
> > > [1] W. Hsu, et al., “Hubert: Self-supervised speech representation learning by masked prediction of hidden units,” IEEE/ACM Transactions on Audio, Speech, and Language Processing, vol. 29, pp. 3451–3460, 2021.

---

### Official Review · Reviewer_VVXW · 2024-10-29

**Soundness:** 1
**Presentation:** 2
**Contribution:** 1
**Rating:** 3
**Confidence:** 4

**Summary:**

This paper proposes learning para-linguistic information within a VAE framework by incorporating semantic information from a speech language model. However, there are issues with both the learning objective formulation and the experimental results.

1. The core idea is to learn an autoregressive prior through VAE training. This approach differs from VQ-VAE, where the prior is learned separately after training. However, the authors use the intermediate features directly to train \(\phi\) and then expect \(\phi\) to generate \(z_t^{c}\) for computing divergence. This does not achieve the traditional effect of KL divergence in regularizing the latent space. Furthermore, it diverges from standard approaches where the prior is parameterized independently from the latent features. Effectively, this approach is equivalent to learning the prior after training, which undermines the formulation's purpose.

2. Conditioning on semantic information to learn para-linguistic features is not new and has already been investigated in works like [1]. Additionally, the audio samples presented by the authors are audibly poor and notably worse than those produced by speech synthesis methods that directly utilize pitch features, as demonstrated in [2]. Furthermore, unlike most recent studies, the authors did not provide a web interface for easy access to samples, which makes evaluation cumbersome. If MOS evaluations were already conducted, it would be beneficial for reviewers to have a similar interface for sample evaluation.

References:
[1] Zhang, Xin, et al. "Speechtokenizer: Unified speech tokenizer for speech large language models." arXiv preprint arXiv:2308.16692 (2023).
[2] Polyak, Adam, et al. "Speech resynthesis from discrete disentangled self-supervised representations." arXiv preprint arXiv:2104.00355 (2021).

**Strengths:**

Nothing particularly strong.

**Weaknesses:**

Theoretical problem and poor result. Please refer to point 1 and 2 in my summary.

**Questions:**

Why no demo page? Have you try train the AR prior post training?

---

> ### Author Response · Authors · 2024-11-20
>
> We thank the reviewer for their feedback and appreciate their insights. We believe there may be a misunderstanding regarding our proposed approach, and we hope this response helps clarify it.
>
> > However, the authors use the intermediate features directly to train ($\phi$) and then expect ($\phi$) to generate ($z_t^{c}$) for computing divergence.
>
> Our method does not use pre-determined intermediate features to train the Encoder $\phi$. Instead, the Encoder $\phi$ directly processes the input speech $\mathbf{X}$ and outputs the distribution of variational posterior for the variational features $\mathbf{Z}^c$.
>
> > This does not achieve the traditional effect of KL divergence in regularizing the latent space. Furthermore, it diverges from standard approaches where the prior is parameterized independently from the latent features.
>
> The Encoder $\phi$ is trained to minimize **two losses**: the reconstruction loss $-E_{q_\phi}[\log p_\theta(\mathbf{X}\mid\mathbf{Z})]$, and the KL divergence loss $E_{\mathbf{X}}[D_{KL}(q_{\phi}(\mathbf{Z}\mid\mathbf{X})||p_\psi(\mathbf{Z}))]$. The KL divergence loss **regularizes the variational feature** $\mathbf{Z}^c$ to be more predictable by the autoregressive prior $p_\psi(\mathbf{Z})$, as shown in the $\mathcal{L}^c_{kl}$ term in Equation 4 of the paper.
>
> Specifically, minimizing $D_{KL}(q_\phi(z^c_{t}\mid \mathbf{X})||p_\psi(z^c_{t}\mid \mathbf{Z}_{1:t-1}))$
>
> for the encoder $\phi$ encourages the variational posterior $q_\phi(z^c_{t}\mid \mathbf{X})$ to approximate $p_\psi(z^c_{t}\mid \mathbf{Z}_{1:t-1})$, making the variational posterior easier to predict autoregressively for $\psi$.
>
> While this differs from the standard VAE framework, where the variational posterior is regularized toward a standard normal distribution, **we believe the KL divergence term in our framework achieves a similar regularizing effect.**
>
> Additionally, a similar autoregressive prior modeling approach was seen in S3VAE [1] for a different application, and we will include this reference to strengthen the theoretical foundation of our work.
>
>
> > Have you try train the AR prior post training?
>
> As mentioned above, the encoder is shaped by both the KL divergence loss and the reconstruction loss. Particularly, since the KL divergence loss term involves the **continuously-updated autoregressive prior** $p_\psi(\mathbf{Z})$, **the resulting variational feature $\mathbf{Z}^c$ is inherently influenced by this prior**. Therefore, in our framework, it is not feasible to pre-train the encoder-decoder pair independently of the autoregressive prior and then train the autoregressive prior separately. **The encoder, decoder, and autoregressive prior are interconnected and optimized together.**
>
> [1] Y. Zhu, et al., S3VAE: Self-supervised sequential vae for representation disentanglement and data generation. In IEEE conference on computer vision and pattern recognition, pages 6538–6547, 2020.

---

> ### Author Response · Authors · 2024-11-20
>
> > Conditioning on semantic information to learn para-linguistic features is not new and has already been investigated in works like [1] (SpeechTokenizer).
>
> First, we would like to clarify to the reviewer that the primary goal of this paper is not to extract para-linguistic features. Rather, our goal is to **improve speech continuation naturalness by leveraging VAE to enable better model the para-linguistic aspects for token-based autoregressive models.** Our experimental results support that we achieved our objective.
>
> Our approach differs from SpeechTokenizer in several key aspects. First, while SpeechTokenizer uses a purely discrete tokenization, our model represents speech through **both discrete and continuous features**. Second, unlike SpeechTokenizer, which does not jointly train its encoder/tokenizer with the autoregressive (AR) model, our method **jointly trains the encoder alongside the AR model**. Third, SpeechTokenizer adopts a pipeline similar to AudioLM, where acoustic tokens serve as prediction targets for a non-autoregressive (NAR) decoder. In contrast, our approach focuses on optimizing the input to the AR model while maintaining a fixed decoder capacity, a distinction further elaborated in Section 6. Finally, SpeechTokenizer addresses text-to-speech (TTS) tasks, whereas our work targets speech continuation, highlighting a **fundamental difference in application domains**.
>
>
> > Additionally, the audio samples presented by the authors are audibly poor and notably worse than those produced by speech synthesis methods that directly utilize pitch features, as demonstrated in [2].
>
> The work referenced by the reviewer addresses a different task from ours. Specifically, **their work evaluates re-synthesis from extracted features of speech (e.g., speech tokens, pitch, and duration), whereas our task focuses on speech continuation—generating future speech based on a given speech prompt.** The samples on their demo page (https://speechbot.github.io/resynthesis/) are all **re-syntheses** (including voice conversion, which is re-synthesis from shuffled features), whereas the samples we provided work on speech continuation with autoregressive modeling. It is expected that speech continuation samples sound perceptually worse than re-synthesis samples, as speech continuation is a significantly more challenging task.
>
>
> > Furthermore, unlike most recent studies, the authors did not provide a web interface for easy access to samples, which makes evaluation cumbersome. If MOS evaluations were already conducted, it would be beneficial for reviewers to have a similar interface for sample evaluation.
> > Why no demo page?
>
> We thought that providing all the samples (100 samples per model) used for human evaluations would offer a comprehensive understanding of the synthesis quality for the reviewers. However, we apologize or any inconvenience caused by requiring the reviewer to manually go through the samples. To make this process easier, we have created a demo page, which can be accessed here: https://anonymous.4open.science/w/demo-12D5/ For compatibility, please use the chrome browser to open the demo page.

---

> ### Comment · Reviewer_VVXW · 2024-11-25
>
> Thank you to the authors for their response. To clarify, my criticism is not about using “pre-determined intermediate features”—I made this clear in my original post. My concern lies in using latent features to directly learn the priors, which defeats the purpose of the KL divergence in a VAE.
> In the original VAE and its variants, the prior is parameterized independently of the latent space. This independence is crucial for the KL divergence to function as intended.
>
>
> My question remains: ***how can you regularize the latent space features using a prior when the prior itself is learned based on latent space features?***

---

> ### Author Response · Authors · 2024-11-25
>
> We sincerely thank the reviewer for the clarification and apologize for any misunderstandings. We hope that this response effectively addresses the concerns raised.
>
> First, we would like to clarify that the concept of a VAE with a parameterized autoregressive prior has been explored in **previous works** [1, 2, 3] in image, video, and text generation, albeit with slightly different modeling approaches. Our work adopts this concept for speech continuation and integrates it with discrete token-based speech language models to jointly model various speech attributes, thereby enhancing the naturalness of the synthesis. These prior works can offer theoretical **grounding for this concept**.
>
> Additionally, we would like to highlight that **we have derived the ELBO formulation for the VAE with an autoregressive prior in Appendix A.1**. This demonstrates that, even with a parameterized prior, maximizing the ELBO continues to **maximize the evidence (log-likelihood)**, which remains the central objective of **generative modeling**. This also provides theoretical grounding for this approach.
>
> **Given the above clarification, we would like to answer the reviewer's question: "how can this framework regularize the latent space features using a prior when the prior itself is learned based on latent space features?"**
>
> We agree with the reviewer that the KL divergence in the original VAE regularizes the latent features toward a standard Gaussian. However, in **generative modeling**, we believe that the primary role of KL divergence is to facilitate **accessible sampling**. In the original VAE, the KL divergence pushes the variational posterior toward a standard Normal prior, enabling sampling from the Normal prior during inference instead of the variational posterior. The regularization effect arises because the latent distribution that achieves the best reconstruction may not naturally align with a Normal distribution. As a result, the latent distribution adjusts by trading off some reconstruction fidelity to better approximate a Normal distribution.
>
> Similarly, in VAE with an autoregressive prior, the KL divergence serves the same purpose by **allowing sampling from a tractable autoregressive prior** as it moves closer to the variational posterior. There is also a regularization effect, as **the latent distribution that provides the best reconstruction may not be perfectly modeled by the parameterized autoregressive prior** due to the limited capacity of the autoregressive model. As a result, the latent distribution also needs to adjust by trading off some reconstruction fidelity to better minimize the KL divergence.
>
> [1] A. Vahdat and J. Kautz, NVAE: A Deep Hierarchical Variational Autoencoder, Advances in Neural Information Processing Systems (NeurIPS), 2020.
>
> [2] Y. Zhu, et al., S3VAE: Self-supervised sequential vae for representation disentanglement and data generation. In IEEE conference on computer vision and pattern recognition, pages 6538–6547, 2020.
>
> [3] X. Fang, et al., Discrete Auto-regressive Variational Attention Models for Text Modeling, 2021 International Joint Conference on Neural Networks (IJCNN), 2021.

---

> ### Author Response · Authors · 2024-11-30
>
> As the discussion period concludes on December 2, we warmly invite the reviewer to share any remaining concerns or feedback. Your valuable insights are greatly appreciated and will help us further refine our work. If possible, we would be grateful if the reviewer could consider revisiting your score, taking the points we address during the discussion into account.

---

> ### Author Response · Authors · 2024-12-02
>
> We would like to provide a follow-up on our earlier response regarding the VAE with parameterized priors, as we believe this is also related to the reviewer's concern about regularization.
>
> As previously suggested, the key difference from the original VAE lies in whether the prior is parameterized. Specifically, the regularizing KL divergence terms are: $D_{KL}(q_\phi(\mathbf{Z}\mid \mathbf{X}))||p_\psi(\mathbf{Z}))$ v.s. $D_{KL}(q_\phi(\mathbf{Z}\mid \mathbf{X})||p(\mathbf{Z}))$.
> A notable distinction between these two terms is that the parameterized prior is more susceptible to KL Vanishing/Posterior Collapse. This is because $D_{KL}(q_\phi(\mathbf{Z}\mid \mathbf{X}))||p_\psi(\mathbf{Z}))$ can more easily be minimized to zero. For instance, $p_\psi(\mathbf{Z})$ could become any trivial distribution—not necessarily Gaussian—while $q_\phi(\mathbf{Z}\mid \mathbf{X})$ could learn to ignore $\mathbf{X}$ and approximate that distribution. As a result, similar to posterior collapse in VAE, the decoder will ignore $\mathbf{Z}$ as it does not provide any information about $\mathbf{X}$, leading to sub-optimal reconstruction loss. Therefore, **this method is actually more prone to be overly regularized**, as the regularization loss here is more likely to be minimized toward zero.
>
> Regarding this challenge, we highlight that, as mentioned in the last paragraph of Section 3.3, we employ a linear warm-up strategy for $\beta$, inspired by previous studies (referenced in our paper), to effectively mitigate the KL Vanishing/posterior collapse problem. With this strategy, we generally do not observe KL vanishing, even up to $\beta=0.08$ (but in this case the reconstruction is distorted). However, without the linear warm-up, KL vanishing could occur as early as $\beta=0.05$.

---

### Official Review · Reviewer_XBpK · 2024-11-01

**Soundness:** 4
**Presentation:** 4
**Contribution:** 3
**Rating:** 8
**Confidence:** 4

**Summary:**

This paper describes an approach to improve speech LM modeling in a speech sequence completion task.  The central idea is to augment the speech tokens with an additional autoregressive variational input.  This results in improved naturalness and meaningfulness of the responses compared to a baseline and a speech-token + pitch representation.

**Strengths:**

The proposal is well motivated, and builds on prior work on speech representation and variational modeling speech generation.

The results are generally quite strong.  The subjective evals show clear improvements to both meaningfulness and naturalness.  The only regressions are to sWUGGY and sBLIMP objective measures.

While the approach adds overall complexity to the inference call, the number of parameters added appear to be quite low -- only 2M additional params.

**Weaknesses:**

It would be useful to have some confidence measure for the sWUGGY, sBLIMP and Perplexity values.  It is unclear how much the 61.75 -> 60.48 sWUGGY regression means.  Is this statistical noise, or an issue that should be addressed.

While the variational augmentation adds fewer than 1% of parameters, it would be useful to know if this meaningfully adds to inference latency.

**Questions:**

See above -- it would help understand the approach more completely to have an understanding of any latency implications to this model adjustment.

---

> ### Author Response · Authors · 2024-11-20
>
> > It would be useful to have some confidence measure for the sWUGGY, sBLIMP and Perplexity values. It is unclear how much the 61.75 $\rightarrow$ 60.48 sWUGGY regression means. Is this statistical noise, or an issue that should be addressed.
>
> Multiple training runs are needed to provide confidence measures for sWUGGY and sBLIMP. While it is computationally infeasible for us to perform multiple runs of training within the rebuttal period. The observed trends in our performance suggest that they are not due to statistical noise.  For instance, in Table 2, we observe a trend that having more information (*Token-LM* $\rightarrow$ *Token-LM + Pitch* $\rightarrow$ *Token-LM + Variational Features*) consistently decreases both sWUGGY and sBLIMP. Also, in Table 4, there is a trend that a higher value of $\gamma$ results in a lower sWUGGY score. It is unlikely that the score differences are statistical noises. Otherwise, there will be no trend from the pure randomness.
>
> > While the variational augmentation adds fewer than 1\% of parameters, it would be useful to know if this meaningfully adds to inference latency.
>
> We thank the reviewer for their feedback. Here, we provide the inference time comparison of different methods:
>
> | Method          | Latency (s) |
> |-----------------|-------------|
> | *Token-LM*      | 6.701       |
> | *Proposed*      | 6.787       |
>
> We report the latency of generating a 10-second continuation out of a 3-second prompt. We run the inferences on a single L40S GPU with a batch size equal to 1. We generate the 10-second continuation with different prompts 50 times and report the average time elapsed in seconds. As shown in the table, our proposed method results in only a 1.3% latency increase.

---

### Official Review · Reviewer_h7h5 · 2024-11-02

**Soundness:** 3
**Presentation:** 2
**Contribution:** 3
**Rating:** 5
**Confidence:** 3

**Summary:**

This paper introduces a variational approach that automatically learns to encode these continuous speech attributes to enhance the speech tokens, and eliminates the need for manual paralinguistic feature selection and extraction, such as pitch features.

**Strengths:**

1. Employing an encoder to derive continuous features instead of manually crafted paralinguistic features endows the features with greater flexibility and potency.
2. The symbols and formulas within the paper are clearly defined, and comprehensive details, encompassing mathematical derivations and experimental setups, are thoroughly documented in the Appendix.
3. Various advanced technologies are used to enhance the model, such as time-wise normalizing flow, and diffusion decoder (However, the ablation studies are not reported in the paper).

**Weaknesses:**

1. In section 3.2, it is mentioned that "By using these tokens, the model no longer needs to encode as much phonetic information in Z^c, allowing Z^c to focus on other continuous speech attributes. ". To strengthen this argument, it would be more convincing to include some analytical experiments. For instance, demonstrating that Z^c excels in speaker verification or emotion recognition, but performs less effectively in speech recognition, would provide a more nuanced understanding of its capabilities.
2. The descriptions of the evaluation are unclear. The authors should provide a clear explanation of whether the AR model is utilized for each metric, possibly by referring to section 3.1 of the GSLM paper.
3. In the main results, the conclusion, "Speech generated from our proposed approach does not sacrifice meaningfulness compared to speech generated from the baselines.", is not strongly supported by the experiments, particularly when considering the observed declines in sWUGGY and sBLIMP. Despite the authors' speculations, the empirical data does not robustly corroborate this claim.
4. it is very weird that the sWUGGY of (Token + continues features) is worse than that of both Token-LM or continues features-LM. (Proposed vs. Token-LM, Proposed vs. Proposed - tokens, in Table 3).

**Questions:**

1. What is the CER of the ground-truth, which serves as the upper bound for the ASR model?
2. When utilizing discrete tokens enriched with acoustic information, such as Encodec, can the proposed method yield enhancements?
3. Why does the Token-LM trained on the LibriSpeech dataset exhibit significantly better CER compared to the Token-LM trained on the Libri-light dataset (5.40 vs. 10.19), yet this improvement is not obtained in the proposed method (5.06 vs. 4.35)?
4. why the numbers of γ=0.5 in Table 4 can not be found in Table 3?
5. why the β is set to 0.04 instead of 0.03 in Table 4?

---

> ### Author Response · Authors · 2024-11-20
>
> > In section 3.2, it is mentioned that "By using these tokens, the model no longer needs to encode as much phonetic information in $Z^c$, allowing $Z^c$ to focus on other continuous speech attributes. ". To strengthen this argument, it would be more convincing to include some analytical experiments. For instance, demonstrating that $Z^c$ excels in speaker verification or emotion recognition, but performs less effectively in speech recognition, would provide a more nuanced understanding of its capabilities.
>
> We thank the reviewer for their valuable suggestions to strengthen our claim, which aligns with similar feedback from Reviewer kqQ1. In the **general response above**, we provide details of the additional experiments conducted. **These analytical experiments provide a more nuanced understanding of the features**, showing that $Z^c$ perform well on emotion recognition, while the utterance embeddings excel in speaker identification tasks. We hope that this addresses the reviewer's feedback.
>
> > The descriptions of the evaluation are unclear. The authors should provide a clear explanation of whether the AR model is utilized for each metric, possibly by referring to section 3.1 of the GSLM paper.
>
> We thank the reviewer for the feedback. In the paper, we evaluate both the reconstruction and continuation capabilities of our model. The AR component is only utilized for continuation. $F_0$-RMSE, MCD, and CER are used to evaluate reconstruction performance, while all other metrics are used to evaluate continuation performance. We update the paper to clarify this distinction as follows:
>  - Similar to the GSLM paper, we will **add a paragraph in Section 4.3** stating that we are evaluating both speech reconstruction and speech continuation, where the reconstruction part doesn't involve the AR model.
>  - We **move N-MOS from Table 1 to Table 2** since the **N-MOS is evaluated on speech continuation**, not speech reconstruction. As a result, Table 1 includes only reconstruction measures that don't involve the AR model,  and Table 2 includes only measures that involve the AR model. We will modify the captions correspondingly for better clarification.
>
> > In the main results, the conclusion, "Speech generated from our proposed approach does not sacrifice meaningfulness compared to speech generated from the baselines.", is not strongly supported by the experiments, particularly when considering the observed declines in sWUGGY and sBLIMP. Despite the authors' speculations, the empirical data does not robustly corroborate this claim.
>
> We thank the reviewer for their thoughtful feedback. M-MOS measures subjective meaningfulness, while sWUGGY and sBLIMP measure objective syntactic and grammatical correctness. We believe that M-MOS is the most direct measure of meaningfulness compared to sWUGGY and sBLIMP. Therefore, we support our claim with the observation that our method achieves comparable (and slightly better) M-MOS scores than other methods. We will revise the text to address the feedback. Specifically stating:  **Speech generated using our proposed approach maintains subjective meaningfulness (as measured by M-MOS) compared to that of the baselines, despite a slight reduction in the objective sWUGGY and sBLIMP scores.**
>
> > It is very weird that the sWUGGY of (Token + continues features) is worse than that of both Token-LM or continues features-LM. (Proposed vs. Token-LM, Proposed vs. Proposed - tokens, in Table 3).
>
> From Table 2, we observe a trend: the **more information encoded in the extracted features, the lower the sWUGGY and sBLIMP scores**. This observation makes us believe that this is the cause of the sWUGGY of *Proposed* is lower than the other methods, as it contains the most information (both tokens and the continuous features). A similar trend is evident in Table 7 in Appendix F, where increasing the number of tokens from $k=200$ to $k=1000$ results in a noticeable decline in both sWUGGY and sBLIMP scores. This supports the idea that the amount of encoded information inversely impacts these metrics.

---

> > ### Author Response · Authors · 2024-11-20
> >
> > > What is the CER of the ground-truth, which serves as the upper bound for the ASR model?
> >
> > The CER of the ground truth is 2.35; we will include the number in Table 1.
> >
> > > When utilizing discrete tokens enriched with acoustic information, such as Encodec, can the proposed method yield enhancements?
> >
> > - We believe that our proposed method can **still offer improvements** since the additional variational features are **optimized end-to-end with the autoregressive model**, which provides a way to shape the extracted representations according to the autoregressive model. On the other hand, the acoustic tokens are trained beforehand and are unaware of the autoregressive modeling task.
> > - In addition, we would like to mention that acoustic tokens encode less structured phonetic information compared to SSL speech tokens. So we believe that using them in replacement of semantic tokens will lead to worse language capability for the autoregressive model.
> >
> > > Why does the Token-LM trained on the LibriSpeech dataset exhibit significantly better CER compared to the Token-LM trained on the Libri-light dataset (5.40 vs. 10.19), yet this improvement is not obtained in the proposed method (5.06 vs. 4.35)?
> >
> > - We thank the reviewer for the thorough review. This observation highlights **a potential advantage of our approach**. In the case of Token-LM, the CER for reconstruction depends entirely on the quality of the extracted speech tokens, as these tokens are the sole input features for the decoder. If the extracted speech tokens lack sufficient phonetic information, the decoder will be unable to reconstruct accurate phonetic content, even with additional training data. Since we evaluate reconstruction on the LibriSpeech-dev dataset, it is reasonable to expect that a speech tokenizer trained on Libri-light (using both k-means and HuBERT) would perform worse than one trained on LibriSpeech, resulting in a higher CER (degradation from 5.40 to 10.19). In contrast, our approach mitigates this issue. Even when the speech tokens are noisier, our encoder can directly learn and extract the necessary information from the input speech. Our approach leverages the extensive data from Libri-light to improve the generalizability of both the encoder and the decoder, leading to better performance (with a CER improvement from 5.06 to 4.35). We will **polish and incorporate this analysis** into the main text of the paper.
> >
> > > Why the numbers of $\gamma=0.5$ in Table 4 can not be found in Table 3?
> >
> > We apologize for the **typo in the caption of Table 4**, which caused this confusion. Models with varying $\gamma$ values were trained on **Libri-light instead of LibriSpeech**. Therefore, the $\gamma=0.5$ in Table 4 is the same setup as the ``Proposed'' setup in Tables 1 and 2. We will update the caption accordingly.
> >
> > > Why the $\beta$ is set to 0.04 instead of 0.03 in Table 4?
> >
> > For computational feasibility, we fix $\beta$ and vary $\gamma$. We choose $\beta=0.04$ as it provides a balance between sWUGGY/sBLIMP scores and reconstruction measures $F_0$-RMSE/MCD/CER as we observed in Table 3.

---

> > > ### Comment · Reviewer_h7h5 · 2024-11-25
> > >
> > > In Table 3, the CER of $\beta$ =0.04 is much worse than $\beta$ =0.03 (20.22% relative), while sWUGGY is only 2.3% relative better.

---

> > > > ### Author Response · Authors · 2024-11-25
> > > >
> > > > We again appreciate the reviewer’s thoughtful feedback and hope this response effectively addresses the concerns.
> > > >
> > > > > sWUGGY and sBLIMP do not support our conclusions.
> > > >
> > > > We agree that sWUGGY and sBLIMP do not directly support our claim about maintaining meaningfulness. However, we also would like to clarify that these metrics also do not contradict our conclusion. While sWUGGY and sBLIMP measure the likelihood of generating real words and grammatically correct sentences, they do not necessarily reflect the meaningfulness perceived by human listeners. We emphasize that M-MOS and N-MOS provide the most relevant evidence for our conclusion, as they directly assess subjective meaningfulness and naturalness. Our results indicate that **our approach preserves subjective meaningfulness, even if it has a higher chance of syntactic or grammatical errors according to objective metrics.** To address the reviewer’s concerns and avoid any confusion, we will clarify the conclusion and main text to clearly explain the distinctions between sWUGGY/sBLIMP and M-MOS, as outlined above.
> > > >
> > > > > The trend of sWUGGY from $k=50$ to $k=200$ is opposite to that of from $k=200$ to $k=1000$
> > > >
> > > > We apologize for overlooking this case in our previous response, and we sincerely thank the reviewer for highlighting it.
> > > > Clearly, this trend does not hold universally—otherwise, having no information in the extracted features would yield the highest scores. Instead, we observed that when working with speech tokens at $k=200$, increasing the encoded information tends to lower the sWUGGY and sBLIMP scores. A reasonable hypothesis is that at $k=50$, the tokens may lack sufficient phonetic detail to perform the two tasks, leading to lower scores. Conversely, encoding excessive information (e.g., at $k=1000$) could introduce unnecessary noise, which also degrades performance. For instance, factors like speech loudness may have low correlation with phonetic content and, when included, add variability that can negatively affect likelihood estimation tasks such as sWUGGY and sBLIMP.
> > > >
> > > > > In Table 3, the CER of  =0.04 is much worse than  =0.03 (20.22% relative), while sWUGGY is only 2.3% relative better.
> > > >
> > > > We appreciate the reviewer’s insightful comments. Comparing relative improvements across different metrics is indeed challenging, especially with different meaning of values (e.g., random guessing performance in sWUGGY is already 50%, which is a high base value).
> > > > As the reviewer suggested, trying $\beta=0.03$ could potentially optimize performance further. However, since our aim was not to optimize performance fully by sweeping $\beta$ but to explore performance variations with different $\gamma$ values, so we chose a balanced $\beta$ of 0.04. We would like to mention that even if this choice is suboptimal, our approach still outperforms the baselines.

---

> > > > > ### Comment · Reviewer_h7h5 · 2024-12-02
> > > > >
> > > > > I would like to thank the author(s) for responding to my questions.
> > > > > I would like to keep my score as is.

---

> > ### Comment · Reviewer_h7h5 · 2024-11-25
> >
> > Thanks for your response, and additional experimental results.
> > However, "the more information encoded in the extracted features, the lower the sWUGGY and sBLIMP scores." is concluded by empirical experimental results. In addition, the trend of sWUGGY from **k=50** to **k=200** is opposite to that of from **k=200** to **k=1000**?
> > In my opinion, these two indicators don't help the authors support their conclusions very well, compared to M-MOS.

---

> ### Author Response · Authors · 2024-11-30
>
> As the discussion period concludes on December 2, we warmly invite the reviewer to share any remaining concerns or feedback. Your valuable insights are greatly appreciated and will help us further refine our work. If possible, we would be grateful if the reviewer could consider revisiting your score, taking the points we address during the discussion into account.

---

### Author Response · Authors · 2024-11-20
**General Response: Additional Experiments**

We sincerely appreciate the reviewers' valuable comments and feedback. In response to their suggestions, we conducted two additional experiments—emotion recognition and speaker identification—to better demonstrate the information encapsulated by our learned features. We will summarize and include these results in the paper.

 | Method                          | Emotion Recognition (ACC, %) |
|---------------------------------|------------------------------|
| *Tokens*                        | 57.46 ± 1.59                |
| *Variational Features*          | 91.57 ± 0.35                |
| *Tokens + Variational Features* | **92.74** ± 0.37            |



| Method                          | Speaker Identification (ACC, %) |
|---------------------------------|----------------------------------|
| *Tokens*                        | 7.08 ± 0.40                    |
| *Variational Features*          | 63.41 ± 0.43                   |
| *Tokens + Variational Features* | 63.13 ± 0.45                   |
| *Utterance Embedding*           | **94.06** ± 0.32               |



- **Speech Emotion Recognition.** We evaluate speech emotion recognition on the EmoV-DB [1] dataset with a single speaker (Sam). We follow a 9:1 split on training and testing for the dataset. The dataset contains 5 emotion categories: amused, angry, neutral, disgust, and sleepiness. We train a classifier with the same structure to predict emotion categories based on different features. The experiments are repeated 20 times to report the mean and 95\% confidence interval. From the above table, we can observe that the variational features alone obtain significantly better performance compared to tokens, showcasing its capability of capturing paralinguistic information. Combining both tokens and variational features gives a slight improvement over using variational features alone.

- **Speaker Identification.** For speaker identification, we evaluate the performance on the VCTK [2] dataset, which consists of read English sentences, with 400 sentences each from 110 speakers. We again follow a 9:1 train-test split and repeat each run 20 times to report the mean and 95\% confidence interval. We additionally evaluate our utterance embedding, which is designed to capture static utterance-level information (see Section 3.5). From the above table, we can see that using tokens only results in poor speaker identification accuracy. With variational features, the classifier obtains improved accuracy. We attribute this improvement to the fact that speaking styles can be captured in the variational features to classify speakers. On the other hand, the utterance embedding outperforms the other features in this task. These results support our claim that the utterance encoder encodes global speaker information while variational features capture local paralinguistic attributes.


[1] A. Adigwe, N. Tits, K.E. Haddad, S. Ostadabbas, and T. Dutoit, The Emotional Voices Database: Towards Controlling the Emotion Dimension in Voice Generation Systems. ArXiv, abs/1806.09514, 2018.

[2] J. Yamagishi, C. Veaux, and K. MacDonald, CSTR VCTK Corpus: English multi-speaker corpus for CSTR voice cloning toolkit (version 0.92), 2019.

---

### Author Response · Authors · 2024-11-28
**Summary of Revisions and Reviewers Feedback Responses**

We sincerely appreciate the reviewers' valuable comments and feedback. As the revision deadline has passed, we would like to remind the reviewers that the paper has been revised to address the raised concerns, with the **updated sections marked in blue** for easy reference.

In summary, the revisions mainly address the following key points:

- **Additional Experiments**: We conducted supplementary experiments to gain a deeper understanding of the variational features, as suggested by Reviewers h7h5 and kqQ1. We included the experiment results in Appendix I.
- **Evaluation Metrics Clarification**: Tables 1 and 2 have been restructured, and we added a paragraph explaining the differences between evaluating speech reconstruction and speech continuation to improve clarity, addressing Reviewer h7h5’s concern.
- **sWUGGY and sBLIMP Clarification**: To address Reviewer h7h5's concern regarding the objective metrics, we clarified that sWUGGY and sBLIMP evaluate syntactic and grammatical correctness, while our claim about meaningfulness is supported by human evaluation scores, which offer the most direct measure. We have revised Section 5.1 to reflect this clarification. Additionally, we hypothesized potential causes for the decline in sWUGGY and sBLIMP scores in Section 5.1 and elaborated further in our author response.
- **Theoretical Background**: To address Reviewer VVXW's concern about the validity of the *VAE with autoregressive prior*, we added references to previous works that adopted parameterized autoregressive prior for VAE in Section 3.1. In our response, we also highlighted our proof that the objective still maximizes ELBO and explained that the regularization effect arises from the autoregressive model's limited capacity.
- We also included a discussion on the conditional independence of $z_t^c$ and $z_t^d$​ in Appendix J and outlined future work on foundation models in Section 8, based on our discussion with Reviewer kqQ1.

We believe these revisions and our responses address the reviewers’ concerns up to this point. However, should any issues remain or further clarifications be needed, **we kindly invite the reviewers to continue the discussion**.

---

### Author Response · Authors · 2024-12-03
**Discussion Period Summary**

We appreciate the positive feedback from the reviewers and their thoughtful evaluation of our work.
 - Reviewer `h7h5` noted that the symbols and formulas are clearly defined and that comprehensive details are thoroughly documented in the Appendix.
 - Reviewer `kqQ1` mentioned that the paper is well-written and provides adequate derivations.
 - Reviewer `XBpK` highlighted that our method is well-motivated and that the results are quite strong.

## Addressing Reviewers’ Concerns:

### Reviewer `kqQ1`:

 - **Additional Experiments**: We conducted supplementary experiments, as suggested, to further explore the variational features. These results are presented in Appendix I, addressing the need for deeper insights and validating our approach. We greatly appreciate the reviewer considering our responses and adjusting the score accordingly.

### Reviewer `XBpK`:

- **Inference speed**: The concern was whether the 1% parameter size increase translates to similar inference latency. We provided a latency comparison, demonstrating only a 1.3% increase, validating the efficiency of our method.

### Reviewer `h7h5`:

- **Clarification for the Evaluation Results**: Reviewer `h7h5` raised concerns about the clarity of presentation. We addressed concerns about presentation clarity by distinguishing between speech reconstruction and speech continuation, restructuring Tables 1 and 2, and adding detailed explanations.
- **sWUGGY and sBLIMP scores**: Reviewer `h7h5` raised concerns about objective metrics. We explained that these metrics evaluate syntactic and grammatical correctness, while our primary claim on meaningfulness relies on human evaluation. We also hypothesized reasons for the trend of the objective scores.

### Reviewer `VVXW`:

 - **Misunderstanding of the Performance Issue**: Reviewer `VVXW` noted that our provided syntheses appear notably worse than those in a prior work. However, as we clarified in our response, the mentioned prior work addresses a re-synthesis task, whereas our focus is on the more challenging problem: speech continuation. We believe this distinction indicates that the comparison is not directly fair and it does not indicate that our method has poor performance.
 - **Questions regarding Theoretical Validity**: Reviewer `VVXW` questioned the validity of the "VAE with auto-regressive prior", specifically about how the latent variables can be regularized with the parameterized prior. In our response, we explained that this approach has been established in prior works and that we derived the corresponding ELBO in Appendix A.1 to provide a solid theoretical foundation. Additionally, we highlighted that the KL divergence term in our formulation continues to offer a regularization effect on the latent space.
 - **Given that we have addressed all main concerns for the initial rejection decision of Reviewer `VVXW`, we believe our paper now merits a higher evaluation.**

---

### Meta-Review · Area_Chair_eZJs · 2024-12-08

**Metareview:**

This paper proposes a variational approach to speech-language modelling compared to traditional auto-regressive models. The central idea is to augment the speech tokens with an additional autoregressive variational input. This results in improved naturalness of the responses compared to a baseline and a speech-token + pitch representation.

The concerns raised by reviewers were mostly addressed during the rebuttal. However, the AC still considers that the paper does not provide sufficient results to demonstrate the usefulness of the proposed approach. Hybrid decoding methods combining semantic and acoustic units are widely used today to incorporate the paralinguistic information of spoken LLMs. However, the paper does not compare its approach with any of these methods. I encourage the authors to benchmark their model against more current approaches.

Please refer to "Additional Comments on Reviewer Discussion" for more details.

**Additional Comments On Reviewer Discussion:**

Reviewer h7h5 considers that the claim in the paper is not strongly supported by the experiments, particularly in light of the observed declines in sWUGGY and sBLIMP. Since the evaluation focuses on syntactic and grammatical correctness, the lack of improvement from the proposed approach is reasonable. The authors will revise the claim to align with these observations.

Reviewer XBpK raised concerns about the inference speed. In response, the authors provided a latency comparison.

Reviewer VVXW questioned the validity of the "VAE with auto-regressive prior," specifically how the latent variables are regularized with the parameterized prior. The authors' rebuttal addressed the concern.

Reviewer VVXW also noted that the provided syntheses appear significantly worse than those in prior work. However, the prior work focuses on a re-synthesis task, whereas this paper is about speech continuation. It is reasonable that continuation performance would be inferior to re-synthesis in this context.


Reviewer kqQ1 questioned the real-world applicability of the approach in tasks like ASR, emotion recognition, or speaker identification. In response, the authors incorporated evaluations on emotion recognition and speaker identification. Since the approach is designed to enhance paralinguistic information, it is understandable that it may not benefit ASR.


Main comments from AC:
The paper proposes a new idea, but the current results reported are insufficient to demonstrate the usefulness of the approach. First, the model performs worse on evaluations focused on syntactic and grammatical correctness. While the proposed approach aims to improve paralinguistic information and may not enhance content-related metrics, the degradation of semantic coherence in favour of natural synthesis is not preferred. Second, more baselines are necessary to establish the utility of the proposed approach. Hybrid decoding methods combining semantic and acoustic units are widely used today to incorporate the paralinguistic information of spoken LLMs. Still, the paper does not compare their approach with any of these methods. I encourage the authors to benchmark their model against more current approaches.

Minor comments from AC:
In the abstract, the authors mention: "These speech tokens typically focus on the linguistic aspects of speech and neglect its paralinguistic content." -> This claim is too assertive. Different speech tokens focus on various aspects of speech; not all focus on content.

---

### Decision · Program_Chairs · 2025-01-22

Reject